# RealtimeTool: Parallel Decoding for Real-Time LLM Function Calling

**Xiaoxin Shi** [1 2]  **Jiaxin Wan** [2]  **Linkang Dong** [2]  **Wei Jiang** [2]  **Yue Liu** [2]  **Zengfeng Huang** [2 3]

 **GitHub:** https://github.com/HaxxorCialtion/SimpleTool
 **Hugging Face:** https://huggingface.co/Cialtion/SimpleTool
 **ModelScope:** https://www.modelscope.cn/models/cialtion/SimpleTool

## Abstract

LLM-based function calling enables intelligent agents to interact with external tools and environments, yet autoregressive decoding imposes a fundamental latency bottleneck that limits real-time applications such as embodied intelligence, game AI, and interactive avatars (e.g., 10 Hz control frequency). We observe that function calling differs fundamentally from free-form text generation: structured outputs exhibit substantial token redundancy (delimiters, parameter names) and weak causal dependencies among arguments—two properties that must be exploited jointly to achieve real-time performance. We present RealtimeTool, which introduces special tokens that serve a dual role: compressing low-entropy tokens ($4$–$6\times$ reduction) while acting as mode selectors that enable independent parallel generation of function name and arguments. This synergistic design achieves $3$–$6\times$ end-to-end speedup (up to $9.6\times$) with only $+8.2\%$ parallelization overhead, while maintaining competitive or improved accuracy across five benchmarks on Qwen-series models (0.5B–14B). With quantization on a consumer-grade GPU, RealtimeTool reaches 61.2 ms P50 latency at 4B scale—enabling 16 Hz real-time control and bridging the gap between LLM function calling and latency-critical real-world deployment.

---

[1]Shanghai Jiao Tong University, Shanghai, China [2]Shanghai Innovation Institute, Shanghai, China [3]Fudan University, Shanghai, China. Correspondence to: Xiaoxin Shi <cialtion737410@sjtu.edu.cn>, Zengfeng Huang <huangzf@fudan.edu.cn>.

*Proceedings of the $43^{rd}$ International Conference on Machine Learning*, Seoul, South Korea. PMLR 306, 2026. Copyright 2026 by the author(s).

## 1. Introduction

Large Language Models (LLMs) have enabled function calling—the mechanism by which LLMs interact with external tools and environments (Brown et al., 2020; O'Neill et al., 2024; Schick et al., 2023). However, a fundamental limitation persists: autoregressive decoding imposes a per-token latency floor that scales linearly with output length (Leviathan et al., 2023; Cai et al., 2024). While streaming masks this latency for conversational applications, function calling requires **complete and valid output before execution**—a partial function call is semantically meaningless. This creates an end-to-end latency bottleneck that streaming cannot alleviate.

### 1.1. The Real-Time Gap

This bottleneck creates a significant gap between LLM capabilities and real-world deployment requirements (Figure 1a).

**Embodied AI & Game AI.** Real-time control demands 5–30 Hz response frequencies, placing the per-action latency budget below roughly 100 ms; autonomous driving pipelines target end-to-end latencies on this same order (Lin et al., 2018), and vision-language-action control loops require at least 10 Hz for stable closed-loop behavior (Yu et al., 2025). Vision-language-action models like OpenVLA (Kim et al., 2025) achieve only 6 Hz at 166 ms latency (Yu et al., 2025)—far below this sub-100 ms threshold for responsive control. In these settings the function call *is* the entire model output and must complete within a single control cycle, so generation latency directly determines whether an LLM-based controller is deployable at all. Game AI systems like Lumine (Tan et al., 2025) achieve 5 Hz only through extensive datacenter infrastructure.

**Edge Deployment.** Mobile agents (Yang et al., 2025; Wang et al., 2024b) rely on cloud APIs with 500 ms–2 s latency per action. Google's FunctionGemma (Google AI, 2025)—a 270M model for edge function calling—trades capacity for speed, underscoring the need for efficient function calling methods.

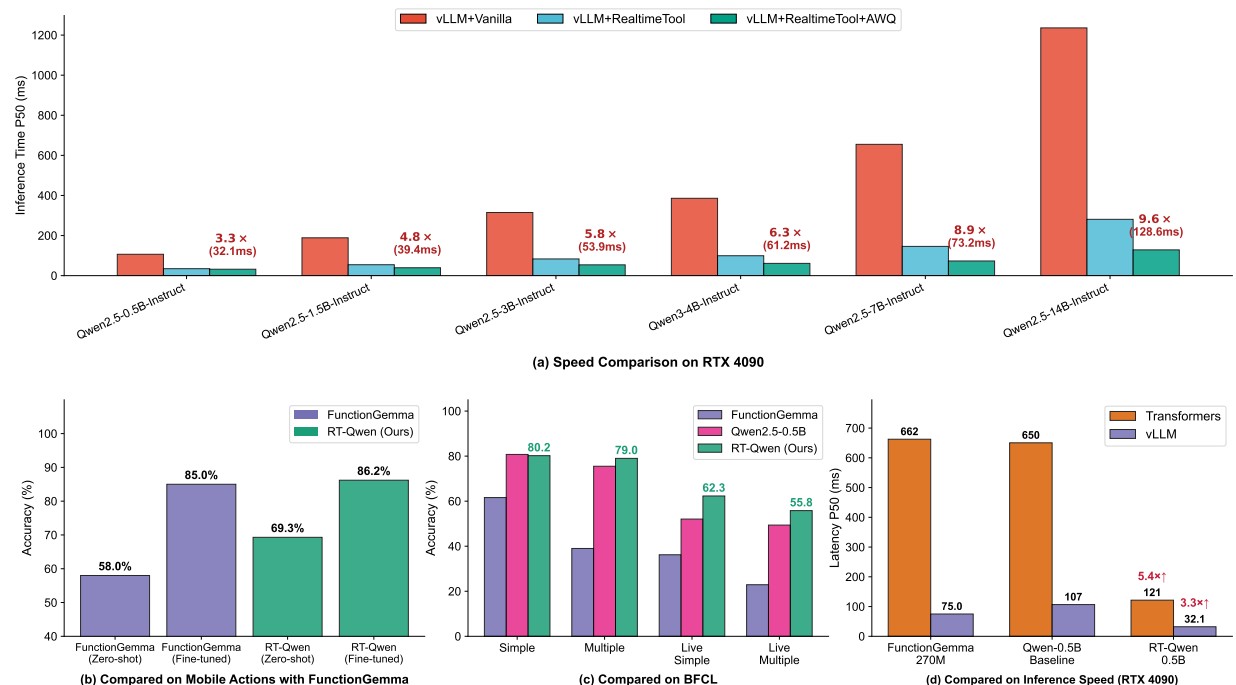

*Figure 1.* Performance Overview of RealtimeTool. **(a)** End-to-end inference latency comparison across various Qwen model sizes on an RTX 4090. Our method achieves up to a **9.6×** speedup. **(b)** Accuracy comparison against FunctionGemma on the Mobile Actions benchmark. **(c)** Accuracy comparison on the BFCL benchmark among FunctionGemma, the Qwen2.5-0.5B baseline, and our RT-Qwen-0.5B. **(d)** Inference latency comparison across different models and frameworks (Transformers vs. vLLM). Latency measurements were conducted on an RTX 4090 using vLLM with prefix caching enabled for the BFCL-v3 benchmark evaluation. Accuracy results for FunctionGemma are sourced from its official model card; all other results are obtained from our evaluation.

### 1.2. Rethinking Acceleration for Function Calling

Function calling end-to-end latency determines the upper bound of LLM-based system capabilities, making acceleration critical. However, existing acceleration methods are not designed for this setting.

**Speculative and Parallel Decoding.** Methods like Medusa (Cai et al., 2024), EAGLE (Li et al., 2024a), and draft-model approaches (Leviathan et al., 2023) accelerate per-token generation through speculation and verification. Though effective for long-form generation, their acceleration benefits are fundamentally capped in short-output scenarios such as function calls: Leviathan et al. (2023) note that the improvement factor is bounded by the number of generated tokens, and the fixed per-cycle overhead of draft inference and verification is amortized over far fewer tokens than in long-form generation. Multi-token prediction (Gloeckle et al., 2024) faces the same output-length ceiling at inference. We take a more fundamental approach: exploiting the memory-bandwidth bottleneck and idle compute capacity during decoding, combined with the inherent weak causal dependencies in function calling, to enable concurrent decoding of function name and arguments—achieving aggressive end-to-end acceleration. Crucially, these approaches are **orthogonal** to ours and can be combined: we validate this

in Appendix M, showing that speculative decoding achieves up to 3.24× additional forward-pass reduction with >95% token acceptance rate when applied to RealtimeTool models, suggesting a combined theoretical speedup of ∼14.6× over vanilla autoregressive decoding.

**Constrained Decoding.** Grammar-constrained frameworks such as Outlines (Willard & Louf, 2023) and XGrammar (Dong et al., 2024) ensure output validity through logits masking. Such masking guarantees well-formed outputs but does *not* reduce the number of forward passes, and the masking step itself adds per-token overhead. The mechanism that does reduce forward passes is jump-forward decoding (e.g., in SGLang (Zheng et al., 2024)), which skips deterministic positions in the output. However, even at its theoretical optimum, jump-forward only removes structurally predictable tokens and remains fundamentally autoregressive; on BFCL-v3 its forward-pass reduction is far smaller than ours (1.38× vs. 4.32×, Appendix J), because it does not address the core bottleneck of token-by-token decoding of high-entropy values.

**Quantization & Compression.** Model compression techniques (Dettmers et al., 2022; Frantar et al., 2023; Lin et al., 2025) provide general acceleration applicable to any task. Our method combines naturally with these approaches (Ta-

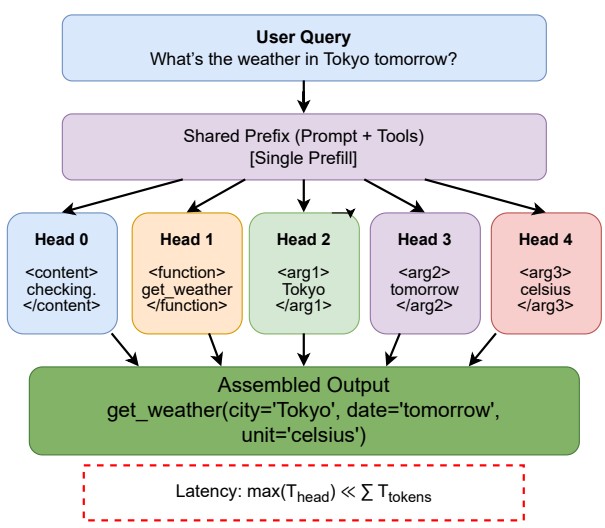

*Figure 2.* Overview of RealtimeTool. Given an input prompt with tool definitions, parallel heads generate function name and arguments independently while sharing the prefix KV cache.

ble 3).

These methods make significant improvements in various aspects. However, they still cannot enable LLMs to make function calls in real time. To achieve this, we advocate a more radical approach that not only leverages parallel decoding but also drastically reduces the number of tokens to be decoded. Furthermore, these two aspects are mutually reinforcing.

### 1.3. Key Insight and Our Approach

We observe that function calls generated by LLMs—typically in JSON or Python API format—exhibit two key properties: weak causal dependencies among arguments and substantial token redundancy. Based on this insight, we propose **RealtimeTool**, which achieves real-time function calling through two synergistic mechanisms (Figure 2):

**Special Tokens.** We introduce special tokens specifically designed for function calling, which are injected into the vocabulary and can effectively compress redundant tokens. Figure 3 illustrates this mechanism. We demonstrate the effectiveness of special tokens in Table 1: **4–6× reduction** compared with vanilla JSON output on samples where both models produce correct outputs. Moreover, these special tokens facilitate parallel decoding as described below.

**Parallel Decoding.** Special tokens further weaken the causal dependencies inherent in function calling, allowing us to treat the function name and its arguments as independent streams. To achieve ultimate speedup, we decode the function name and arguments in parallel. These parallel streams share the same input prefix and differ only in the

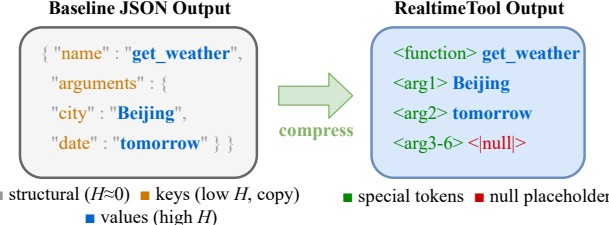

*Figure 3.* Token compression illustration. Baseline structured output contains ∼30 tokens spanning three entropy levels. RealtimeTool compresses low-entropy tokens into special markers, retaining only high-entropy values for generation.

appended special token, requiring only a single prefill operation with full reuse of the prefix KV cache. This drastically reduces memory overhead and enables full utilization of idle compute capacity. Consequently, negligible overhead is observed in the time per output token (TPOT) test presented in Appendix G.

### 1.4. Results Overview and Contributions

We conduct comprehensive evaluation on Qwen-series models ranging from 0.5B to 14B parameters across **five benchmarks**: BFCL-v3 (Patil et al., 2025), Mobile Actions (Google AI, 2025), SealTools (Wu et al., 2024), OpenFunction (Patil et al., 2024), and ToolAlpaca (Tang et al., 2023), covering diverse function calling scenarios from API invocation to mobile device control.

Results demonstrate that RealtimeTool achieves **3–6× speedup** with competitive or improved accuracy across all model sizes and benchmarks. Combined with quantization, our method enables sub-100ms function calling latency with 4B-scale models on consumer-grade GPUs, reaching real-time control frequencies previously infeasible with autoregressive decoding. We additionally compare against Google's recent model, FunctionGemma (Google AI, 2025), on the Mobile Actions benchmark. In both zero-shot and fine-tuned settings, our smallest model, RT-Qwen2.5-0.5B, demonstrates advantages in both accuracy and latency for edge deployment scenarios.

**Contributions.**

(1) We propose **RealtimeTool**, a parallel decoding framework that enables LLMs to call functions in real time, making it possible for LLM-based methods to control real-time interactive systems.

(2) We design mutually reinforcing special tokens and a parallel decoding architecture that exploits idle compute capacity, conducting an in-depth exploration of redundant token compression and parallel decoding for function calling.

(3) We train multiple RealtimeTool models across the Qwen series (0.5B–14B) and evaluate them on five benchmarks, demonstrating substantial speedup with competitive accuracy.

(4) We compare against Google's FunctionGemma on Mobile Actions in both zero-shot and fine-tuned settings, showing more effective edge deployment capability.

**Conflict of Interest Disclosure.** We evaluate publicly available models that were not developed by the authors or their affiliated institutions, including the Qwen series, and we compare against Google's FunctionGemma using its publicly reported results and released checkpoints. None of the authors has a financial interest in any of the models or systems evaluated in this work, and the authors declare no financial conflicts of interest.

## 2. Methodology

Building on the insight that function calling exhibits structural redundancy and weak causal dependencies (Section 1), this section presents the technical realization of RealtimeTool. The key challenge is that these two properties must be exploited *jointly*: compressing redundant tokens alone does not break sequential decoding, while parallelizing without compression leaves substantial latency unreduced.

We describe how special tokens and parallel decoding work synergistically: special tokens not only compress output length (4–6× reduction) but also serve as **mode selectors** that enable independent generation of function name and arguments (Section 2.1). The training strategy then ensures the model learns to leverage both mechanisms effectively (Section 2.2).

### 2.1. Special Tokens Enable Parallel Decoding

2.1.1. FROM REDUNDANCY TO COMPRESSION

Intuitively, conventional function call outputs contain three token categories with distinct entropy characteristics. (Figure 3):

- **Structural tokens** ({, }, :, , ): Fully predictable given the output format, near-zero entropy.
- **Key tokens** ("name", "arguments", parameter names): Copied from tool definitions, low entropy.
- **Value tokens** (function names, argument values): Require actual LLM reasoning, high entropy.

We introduce special tokens (`<function>`, `<arg1>`–`<arg6>` e.g.) that absorb all low-entropy information, achieving 4–6× compression (Table 1). Crucially, these tokens serve a dual purpose: beyond compression, each

*Table 1.* Token compression ratio across model sizes. CR = Baseline tokens / RealtimeTool bottleneck-head tokens, computed as $\mathrm{BL}_p/\mathrm{RT}_p$ at each statistic (not the percentile of the per-sample CR distribution). Statistics computed on samples where both models produce correct outputs. Per-benchmark breakdown in Appendix H.

| Model | CR (Mean) | CR (P50) | CR (P90) |
|---|---|---|---|
| Qwen2.5-0.5B | 5.06× | 6.00× | 4.06× |
| Qwen2.5-1.5B | 4.32× | 4.71× | 3.20× |
| Qwen2.5-3B | 4.38× | 4.71× | 3.20× |
| Qwen3-4B | 4.32× | 4.86× | 3.42× |
| Qwen2.5-7B | 5.35× | 5.00× | 3.40× |
| Qwen2.5-14B | 4.51× | 4.86× | 3.47× |
| **Average** | **4.66×** | **5.02×** | **3.46×** |

token acts as a **mode selector**, signaling which component (function name, specific argument or even content) to generate. This dual role—compression *and* mode selection—is what enables parallel decoding.

2.1.2. FROM WEAK DEPENDENCIES TO PARALLEL DECODING

With special tokens serving as mode selectors, we can exploit the weak causal dependencies among function arguments: since each argument can be inferred largely independently from the input context, we decode them as **parallel streams** rather than sequentially.

**Architecture.** As illustrated in Figure 2, RealtimeTool decouples function calling into $H$ independent decoding streams (1 for function name, up to 6 for arguments, more if needed in training), each initialized by appending a distinct special token (`<function>`, `<arg1>`, ..., `<arg6>`) to the shared input prefix. We also reserve a content head (`<content>`) for chatting, which can be optionally used alongside function calls. All streams share the same prefix KV cache; only the head-specific special token differs. This design directly leverages the dual role of special tokens: they compress the output *and* specify which component to generate.

**Latency Analysis.** For conventional autoregressive decoding, end-to-end latency is:

$$T_{\text{baseline}} = T_p + N \cdot T_d \qquad (1)$$

where $T_p$ is prefill time, $N$ is total output tokens, and $T_d$ is per-token decode latency. With RealtimeTool, the latency becomes:

$$T_{\text{ours}} \approx T_p + \max_{i \in \{1,...,H\}} (N_i) \cdot T_d \qquad (2)$$

where $H$ is the number of parallel decoding heads, $N_i$ is the token count for head $i$, and some insignificant overhead may exist. End-to-end latency is thus dominated by the **longest single head**, not the sum of all heads.

### 2.1.3. PARALLEL DECODING AS A NEAR-FREE OPERATION

One might expect $H$ parallel decoding streams to incur proportional computational overhead. We argue—and empirically validate—that this cost is negligible due to the **memory-bandwidth-bound** nature of autoregressive decoding.

During the decode phase, each token generation requires loading the full model weights from GPU memory, while the actual computation (a single matrix-vector product per layer) utilizes only a fraction of available FLOPS. This creates substantial idle compute capacity. Batching multiple sequences—in our case, parallel heads within the same request—increases arithmetic intensity without proportionally increasing memory traffic, effectively utilizing otherwise-idle compute.

We validate this through batch scaling experiments (shown in Appendix G). Across six model configurations from 0.5B to 14B parameters with the same prefix and one different token, 8-head parallel decoding achieves **93.0% average TPOT efficiency**, with efficiency only dropping markedly at extremely large batch sizes, demonstrating that RealtimeTool's parallel heads operate well within the memory-bound bottleneck and idle compute capacity.

**KV Cache Efficiency.** All parallel heads share the input prefix, enabling full KV cache reuse. For an input of $L$ tokens and $H$ heads each appending a single special token, the KV cache overhead is only $H$ tokens beyond the shared prefix—negligible compared to typical prompt lengths. In high-frequency control scenarios where the tool schema remains fixed, this shared prefix can be persistently cached, further amortizing prefill cost (see Section 3.4 for empirical validation).

### 2.2. Training for Parallel Generation

The synergistic design described above requires the model to learn a fundamentally different output behavior: generating function name and arguments as **independent streams** conditioned on distinct special tokens, while maintaining reasoning capability.

#### 2.2.1. SPECIAL TOKEN DESIGN

We introduce 17 special tokens organized into four functional groups:

**Content head**: `<content>`, `</content>` for natural language responses.
**Function head**: `<function>`, `</function>` for function name generation.
**Argument heads**: `<arg`$k$`>`, `</arg`$k$`>` for $k \in \{1, ..., 6\}$, each selecting a positional argument.

**Placeholder**: `<|null|>` for unused argument slots, enabling early termination of inactive heads.

Each special token encodes all structural information (delimiters, parameter names, positional ordering) into its semantics, eliminating the need to generate them explicitly while simultaneously specifying the generation mode.

#### 2.2.2. TRAINING CHALLENGES

Achieving high-quality parallel decoding is non-trivial. Unlike standard fine-tuning where the model learns a single output distribution, RealtimeTool requires the model to learn **eight distinct output modes** that share the same input but produce semantically different outputs. This poses two key challenges:

**Capacity Requirements.** Learning multiple output modes demands sufficient model plasticity. We find that standard LoRA configurations underperform for this task; significantly larger adapter capacity is required to capture the diverse output patterns without interference between heads (see ablation in Section 3.5). Moreover, the introduction of special tokens necessitates unfreezing the embedding layer—a departure from standard parameter-efficient fine-tuning that requires careful optimization to balance learning new token representations against preserving pretrained knowledge.

**Data Distribution Balance.** Existing function calling datasets exhibit skewed argument count distributions, with most samples concentrated around 2–3 arguments. For RealtimeTool's multi-head architecture, underrepresented argument counts (0, 5, 6 arguments) lead to poorly trained heads that fail to generalize. We address this through targeted data augmentation using synthetic samples that balance argument count distribution (see ablation in Section 3.5).

#### 2.2.3. TRAINING PIPELINE

We adopt parameter-efficient fine-tuning with LoRA ([Hu et al., 2022](#)), targeting the MLP layers to maximize the capacity for learning head-specific output patterns. Each function-calling sample is decomposed into eight parallel sequences—one per head (content, function, and `arg1–arg6`)—that share an identical prefix but differ in the appended special token and the corresponding target output. Each head is trained with a standard next-token prediction objective on its own target, and the per-head losses are combined with head-specific weighting to address output-length imbalance across heads (Appendix F). The model also learns *whether* a tool call is needed, not only *what* to call: we include ∼2% zero-argument pass-through samples (Appendix D) so that the function head can abstain when no tool is appropriate. Because each head is processed as an independent sequence, training incurs roughly 5× the wall-

clock cost of standard single-sequence fine-tuning, with *no* additional GPU memory; this overhead is a one-time model-preparation cost and could be further reduced by prefix sharing during training. Subsequent domain adaptation is far cheaper, as shown in Section 3.4. Full hyperparameters are provided in Appendix F.

# 3. Experiments

We evaluate RealtimeTool from four perspectives: (1) **accuracy** on multiple benchmarks (Section 3.2), (2) **inference speedup** (Section 3.3), (3) **domain adaptation** comparing with FunctionGemma (Section 3.4), and (4) **ablation studies** (Section 3.5).

## 3.1. Experimental Setup

**Models.** We evaluate on Qwen2.5-Instruct series (0.5B–14B) (Qwen Team, 2024) and Qwen3-4B-Instruct (Yang et al., 2025). All models are fine-tuned using LoRA (Hu et al., 2022) with a relatively large rank (we explore this setting in Section 3.5). We set the number of parallel argument heads to 6, which already covers **95.2%** of function calls across all evaluation benchmarks (see Appendix C for detailed statistics).

**Benchmarks.** We evaluate on five benchmarks covering diverse function calling scenarios: BFCL-v3 (Patil et al., 2025) (single-turn subsets),[1] Mobile Actions (Google AI, 2025), SealTools (Wu et al., 2024), OpenFunction-v1 (Patil et al., 2024), and ToolAlpaca (Tang et al., 2023). The last three benchmarks are combined into an "Others" group for aggregate evaluation.

**Metrics.** Overall Accuracy (complete correctness including function name and all arguments) and Function Accuracy (function name selection only).

**Hardware.** Inference latency is measured on RTX 4090 and H100 GPUs.

## 3.2. Accuracy Evaluation

Table 2 and Figure 4 present comprehensive accuracy evaluation across model scales and benchmarks.

**Consistent improvement across model scales.** All model sizes show overall accuracy gains ranging from +0.5% (1.5B) to +3.5% (7B), with an average improvement of +2.9%. This consistency suggests that models have learned these new output modes effectively.

---

[1] We cite two distinct works by overlapping author groups: Patil et al. (2024) (Gorilla, NeurIPS 2024), which introduced the OpenFunction benchmark, and Patil et al. (2025) (the Berkeley Function Calling Leaderboard, ICML 2025), which introduced the BFCL-v3 evaluation protocol. We cite the former for OpenFunction-v1 and the latter for BFCL-v3.

**Substantial function accuracy gains.** Average function accuracy increases +7.1% over baselines, with several configurations achieving near-perfect scores. We attribute this to special tokens acting as mode selectors: by explicitly conditioning each head on its target component, the model avoids ambiguity in output structure. Additionally, the compressed output format means fewer tokens and thus fewer opportunities for errors—particularly important in function calling where a single incorrect token invalidates the entire output.

**Strongest gains on Mobile Actions.** Improvements range from +7.3% to +13.3% across model sizes. Notably, this benchmark was released in December 2025 (Google AI, 2025), after both baseline models and our training data were finalized, demonstrating strong generalization to unseen domains.

## 3.3. Speedup Evaluation

Figure 5 presents speedup evaluation in different settings, demonstrating the effectiveness of RealtimeTool's acceleration strategies.

**Basic Parallelization** (Figure 5a): With the Transformers backend, RealtimeTool achieves 2.8–5.4× speedup on RTX 4090 and 3.0–6.4× on H100. Smaller models benefit more due to their decode-bound characteristics.

**Prefix Caching** (Figure 5b): vLLM's KV cache sharing across parallel streams yields 2.7–4.4× speedup, with larger models showing improved ratios as prefix caching amortizes shared computation more effectively.

**Quantization** (Figure 5c): AWQ 4-bit quantization achieves the lowest absolute latency. Qwen3-4B reaches **61.2ms** P50, corresponding to 16 Hz control frequency—exceeding the 10 Hz threshold typically required for real-time applications.

**Parallelization Overhead** (Figure 5d): As the number of active argument heads increases from 1 to 6, speedup remains stable, confirming that parallelization overhead is negligible—consistent with our analysis that parallel streams utilize otherwise-idle compute capacity within the memory-bandwidth-bound decode phase. Note that Figure 5d reflects end-to-end measurements where varying output lengths across heads introduce additional variance; controlled experiments isolating batch size effects are provided in Appendix G.

Table 3 summarizes results across the acceleration hierarchy. Models up to 7B achieve sub-100ms latency on consumer-grade RTX 4090, enabling real-time deployment without datacenter infrastructure.

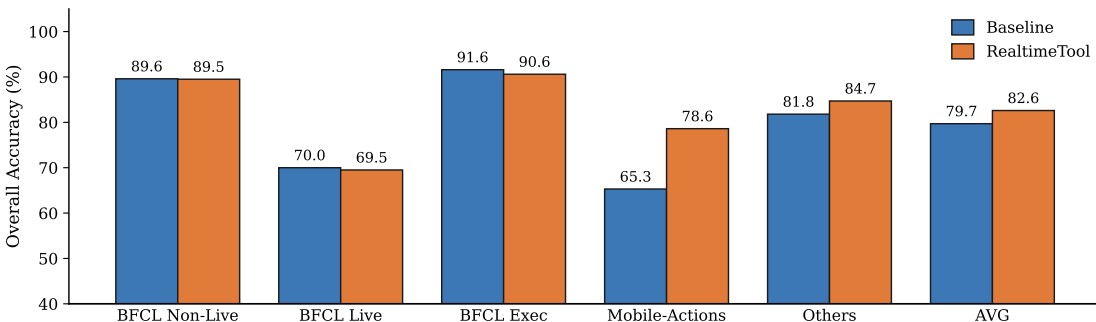

*Figure 4.* Average accuracy comparison between Baseline and RealtimeTool across benchmark groups (macro average). BFCL-v3 is evaluated on single-turn subsets; Mobile Actions parallel calls are converted to multi-turn format; "Others" combines SealTools, OpenFunc, and ToolAlpaca.

*Table 2.* Performance comparison on five benchmarks. Each cell shows overall (function) accuracy, e.g., 78.3 (98.3). Speedup ratio after model name indicates inference acceleration on RTX 4090 with vLLM (P50 latency). **Bold** = best, underline = second best within each column (applied separately to overall and function accuracy).

| Model | BFCL Non-Live | BFCL Live | BFCL Exec | Mobile Actions | Others | AVG |
|---|---|---|---|---|---|---|
| *Baseline Models* | | | | | | |
| Qwen2.5-0.5B (1×) | 78.3 (98.3) | 49.9 (86.4) | 88.6 (96.6) | 56.7 (74.4) | 80.9 (95.2) | 70.9 (90.2) |
| Qwen2.5-1.5B (1×) | 86.3 (98.3) | 67.4 (90.4) | 91.9 (98.0) | 65.5 (74.9) | 82.2 (95.3) | 78.7 (91.4) |
| Qwen2.5-3B (1×) | 92.5 (99.7) | 73.0 (94.3) | 86.6 (96.0) | 66.2 (75.0) | 84.3 (95.9) | 80.5 (92.2) |
| Qwen3-4B (1×) | 93.7 (99.3) | **77.9** (94.5) | **94.6** (97.3) | 68.0 (75.0) | 82.0 (91.8) | 83.2 (91.6) |
| Qwen2.5-7B (1×) | 93.0 (99.7) | 75.4 (95.2) | 94.0 (99.3) | 68.4 (77.2) | 83.4 (94.9) | 82.8 (93.3) |
| Qwen2.5-14B (1×) | **94.0** (99.3) | 76.6 (95.9) | 94.0 (98.7) | 67.0 (75.0) | 78.3 (89.5) | 82.0 (91.7) |
| *RealtimeTool (Ours)* | | | | | | |
| RT-Qwen2.5-0.5B (3.1×) | 79.8 (**99.8**) | 57.2 (91.6) | 87.9 (**100.0**) | 69.3 (99.3) | 82.2 (99.1) | 75.3 (98.0) |
| RT-Qwen2.5-1.5B (3.5×) | 88.8 (**99.8**) | 63.6 (94.3) | 90.6 (**100.0**) | 72.8 (**100.0**) | 80.2 (99.3) | 79.2 (98.7) |
| RT-Qwen2.5-3B (3.8×) | 90.3 (**99.8**) | 68.0 (95.4) | 91.3 (**100.0**) | 81.4 (99.9) | 85.5 (99.5) | 83.3 (98.9) |
| RT-Qwen3-4B (3.9×) | 92.5 (99.5) | 76.4 (96.3) | 89.9 (**100.0**) | **84.5** (99.9) | 86.6 (**99.7**) | 86.0 (99.1) |
| RT-Qwen2.5-7B (4.5×) | 93.5 (**99.8**) | 75.8 (96.7) | 92.6 (**100.0**) | 83.2 (**100.0**) | 86.2 (99.5) | **86.3** (99.2) |
| RT-Qwen2.5-14B (4.4×) | 92.2 (99.7) | 75.9 (**97.1**) | 91.3 (**100.0**) | 80.5 (**100.0**) | **87.4** (**99.7**) | 85.5 (**99.3**) |

*Table 3.* Speedup and latency summary on RTX 4090.

| Model | Trans. | vLLM | AWQ (ms) |
|---|---|---|---|
| Qwen2.5-0.5B | 5.35× | 3.07× | 32.1 |
| Qwen2.5-1.5B | 3.32× | 3.46× | 39.4 |
| Qwen2.5-3B | 2.93× | 3.78× | 53.9 |
| Qwen3-4B | 2.83× | 3.88× | 61.2 |
| Qwen2.5-7B | – | – | 73.2 |
| Qwen2.5-14B | – | – | 128.6 |

*Table 4.* Accuracy on Mobile Actions.

| Model | Zero-shot | Fine-tuned |
|---|---|---|
| Qwen3-4B-Instruct | 83.7 | – |
| FunctionGemma (270M) | 58.0 | 85.0 |
| RT-Qwen-0.5B (Ours) | **69.3** | **86.2** |
| Δ vs FunctionGemma | +11.3 | +1.2 |

### 3.4. Case Study: Comparison with FunctionGemma

We compare against FunctionGemma (Google AI, 2025), Google's 270M model released in December 2025 specifically for edge function calling, to evaluate whether RealtimeTool can compete with industry solutions designed for on-device deployment.

**Setup.** We evaluate on the Mobile Actions dataset released alongside FunctionGemma, containing ∼9,600 samples covering 7 smartphone operations. This benchmark complements BFCL-v3 by focusing on mobile device control—a scenario closely aligned with our target real-time applica-

tions. For domain adaptation, we start from the already-trained RT-Qwen-0.5B and further fine-tune it on the official Mobile Actions training split. Since this step adapts an existing parallel-decoding model rather than learning the eight output modes from scratch, a small adapter suffices: we use LoRA with $r = 64$, requiring less than 3 H100-hours. Latency is measured on the full test split (1,283 samples) using vLLM with prefix caching.

**Results** (Tables 4–5): RT-Qwen-0.5B achieves 69.3% zero-shot (+11.3% vs FunctionGemma) and 86.2% after fine-tuning, surpassing both FunctionGemma (85.0%) and the 8× larger Qwen3-4B (83.7%). Despite 1.8× more parameters, RT-Qwen-0.5B achieves faster inference (51.0ms vs

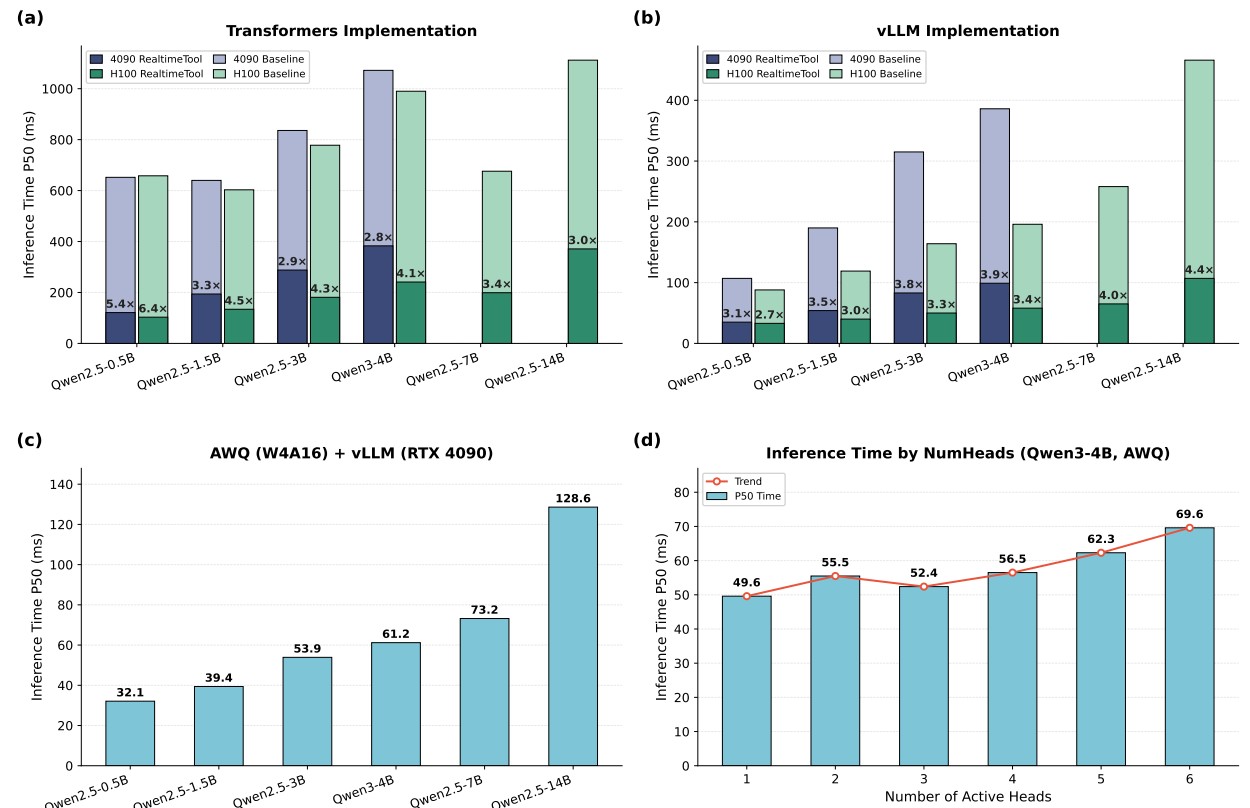

*Figure 5.* **Inference speedup evaluation.** (a)–(b) Stacked bars show baseline time partitioned into RealtimeTool time (dark) and time saved (light); speedup ratios labeled at boundary. (a) Transformers backend on RTX 4090 (blue) and H100 (green). (b) vLLM backend with prefix caching. (c) Absolute latency with AWQ quantization on RTX 4090. (d) Latency scaling with the number of active argument heads (Qwen3-4B, AWQ on RTX 4090); the function head is always active and the bottleneck head is invariably an argument head. All measurements on BFCL-v3 single-turn cases.

*Table 5.* Latency on Mobile Actions (vLLM, RTX 4090).

| Model | P50 (ms) | P90 (ms) | Speedup |
|---|---|---|---|
| Qwen3-4B (Baseline) | 468.5 | 667.0 | 1.0× |
| Qwen2.5-0.5B (Baseline) | 110.1 | 154.0 | 4.3× |
| FunctionGemma (270M) | 61.1 | 139.5 | 7.7× |
| **RT-Qwen-0.5B (Ours)** | **51.0** | **74.5** | **9.2×** |

61.1ms P50) with substantially better tail latency (74.5ms vs 139.5ms P90)—critical for real-time applications where worst-case latency determines reliability. We further validate these gains on genuine edge hardware: in an apples-to-apples on-device iPhone comparison, RT-Qwen-0.5B is faster than FunctionGemma at both P50 and P90 with cooler thermal behavior (Appendix L).

### 3.5. Ablation Studies

We conduct ablation studies on Qwen3-4B evaluated on BFCL-v3 (Table 6).

**LoRA Rank.** The goal of this ablation is to characterize the adapter capacity required to learn eight distinct output modes. A clear capacity floor exists below rank 64: accuracy collapses to 69.2% at rank 16 and 73.7% at rank 32, indicating that small adapters cannot represent the multiple head-specific distributions without destructive interference. Above this floor, accuracy plateaus—from rank 64 to 1024 it varies within roughly two points (85.3–87.0%), with rank 1024 attaining the best score. We adopt rank 512 as the default not because it is the single best setting, but because it lies safely within the plateau while fully utilizing our single-GPU memory budget; rank 1024 strains memory and slows training for a marginal (+1.7%) gain.

*Table 6.* Ablation studies on BFCL-v3 (Qwen3-4B). Left: LoRA rank, showing a capacity floor below 64 and a plateau above it. Right: training-data composition.

| LoRA Rank | Acc. (%) | Training Data | Acc. (%) |
|---|---|---|---|
| 16 | 69.2 | xLAM only (public) | 85.3 |
| 32 | 73.7 | xLAM + Synthetic | **86.3** |
| 64 | 85.5 | | |
| 256 | 86.4 | | |
| 512 | 85.3 | | |
| 1024 | **87.0** | | |

**Training Data.** Using only public xLAM data (Zhang et al., 2025), RealtimeTool achieves 85.3%—demonstrating strong performance without proprietary data. Synthetic augmentation via Qwen3-235B yields +1.0% improvement. We emphasize that this augmentation balances the argument-count distribution for multi-head learning (Section 2.2) rather than adding general task supervision.

**Source of gains: design vs. capacity.** To verify that the accuracy gains stem from the parallel-decoding design rather than from the additional trainable parameters introduced by a high LoRA rank, we train a vanilla supervised fine-tuning (SFT) baseline that produces standard JSON outputs using the *identical* data and LoRA rank ($r = 512$) as RealtimeTool (Table 7). Under matched parameters and data, vanilla SFT *decreases* overall accuracy to 71.3% (while function accuracy rises to 97.0%, confirming that training itself succeeds), whereas RealtimeTool improves overall accuracy to 86.0%. The gain is therefore attributable to the compressed, structurally explicit output format and parallel decoding, not to extra capacity. We further verify in Appendix K that this specialization does *not* degrade general capability: under standard prompting (no special tokens), RT-Qwen3-4B matches the baseline on MMLU and slightly improves on IFEval.

*Table 7.* Isolating the source of gains on BFCL-v3 (Qwen3-4B). All configurations use identical training data; the two trained rows use identical LoRA rank ($r = 512$). These runs use a slightly different training configuration from the main experiments, so absolute numbers are not directly comparable; the relative gap between vanilla SFT and RealtimeTool is unaffected.

| Configuration | Format | Overall | Func |
|---|---|---|---|
| Baseline (no training) | JSON | 83.2 | 91.6 |
| Vanilla SFT ($r$=512) | JSON | 71.3 | 97.0 |
| RealtimeTool ($r$=512) | Parallel | **86.0** | **99.1** |

## 4. Limitations

**Argument Independence.** RealtimeTool assumes function arguments can be generated independently—valid for well-designed APIs with unordered key-value parameters. When semantic dependencies exist (e.g., `file_path` and `line_number`), dependent parameters can be consolidated into a single head, trading parallelism for correctness.

**Fixed Head Count.** The 6-argument head configuration covers 95.2% of functions in our benchmarks. APIs exceeding this limit can concatenate overflow parameters into the final head, preserving functionality with reduced parallelism.

**Training Cost.** Processing each head as an independent sequence incurs computational overhead compared to standard fine-tuning, acceptable for one-time model preparation

but limiting rapid iteration.

## 5. Future Work

**Vision-Language Extension.** Extending RealtimeTool to VLMs would enable real-time Vision-Language-Action systems, with efficient visual token handling as the key challenge.

**On-Device Deployment.** Bridging server-side speedup to edge requires optimization for mobile frameworks (LiteRT, Core ML, ExecuTorch) and device-side adaptation.

**Adaptive Heads.** Dynamically adjusting active head count based on tool schemas would improve flexibility for diverse APIs.

**Scaling.** Whether accuracy gains persist on 30B+ models remains open.

## 6. Conclusion

We presented RealtimeTool, a parallel decoding framework for real-time LLM function calling. Our key insight is that structured function outputs exhibit structural redundancy and weak causal dependencies among arguments—two properties that must be exploited jointly. We introduced special tokens that serve a dual role: compressing predictable tokens (4–6× reduction) while acting as mode selectors that enable independent generation of function name and arguments. This synergistic design allows parallel decoding streams to share the prefix KV cache with negligible overhead, achieving 3–6× end-to-end speedup while maintaining competitive accuracy across five benchmarks. Combined with quantization, RealtimeTool enables 4B-scale models to achieve 16+ Hz real-time function calling on consumer GPUs, bridging the gap between LLM capabilities and latency-critical applications such as embodied AI, game AI, and interactive agents.

## Impact Statement

This paper advances real-time LLM function calling, enabling latency-critical applications such as embodied AI, game AI, and interactive agents. We foresee positive societal impacts including more responsive assistive robotics, accessible AI tutoring systems, and democratized deployment of capable AI on consumer hardware.

Potential risks include misuse in autonomous systems operating without adequate human oversight. We recommend that practitioners deploying RealtimeTool in safety-critical domains implement appropriate safeguards, monitoring mechanisms, and human-in-the-loop controls.

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

# A. Other Related Work

**LLM Function Calling and Tool Use.**    The ability of LLMs to interact with external tools has been extensively studied. Toolformer (Schick et al., 2023) explored self-supervised tool learning, while Gorilla (Patil et al., 2024) and ToolLLM (Qin et al., 2024) improved accuracy through retrieval-augmented approaches. xLAM (Zhang et al., 2025) addresses function calling through data augmentation strategies. At the orchestration level, several works explore parallelism and scheduling: LLM Compiler (Kim et al., 2024) investigates parallel function calling, Skeleton-of-Thought (Ning et al., 2024) explores parallel generation via prompting strategies, AsyncLM (Gim et al., 2024) proposes asynchronous execution patterns, and GhostShell (Gong et al., 2025) addresses concurrent tool execution in embodied settings. LLMA (Yang et al., 2023) explores reference-based acceleration techniques. For evaluation, BFCL (Patil et al., 2024) and StableToolBench (Guo et al., 2024) provide standardized benchmarks, while Mobile Actions (Google AI, 2025) targets edge deployment scenarios. Google's FunctionGemma (Google AI, 2025) represents industry efforts toward compact, edge-deployable models.

**Speculative and Parallel Decoding.**    Speculative decoding (Leviathan et al., 2023; Chen et al., 2023) accelerates inference using draft-and-verify mechanisms. Medusa (Cai et al., 2024) and the EAGLE series (Li et al., 2024a;b; 2025) introduce parallel decoding heads, while Hydra (Ankner et al., 2024) explores sequentially-dependent draft heads. Multi-token prediction (Gloeckle et al., 2024) trains models to predict multiple future tokens. A comprehensive survey is provided by Xia et al. (2024). Unlike these general-purpose methods, RealtimeTool exploits the structural properties of function calling, achieving parallelism without additional parameters or draft models.

**Structured Output Generation.**    Grammar-constrained approaches such as Outlines (Willard & Louf, 2023) and XGrammar (Dong et al., 2024) enforce output validity through logits masking, which guarantees well-formed structure but does not reduce the number of forward passes. The complementary technique that does reduce forward passes is jump-forward decoding, as used in SGLang (Zheng et al., 2024), which emits deterministic spans without a model call. These methods improve validity and skip predictable delimiters, but they remain fundamentally sequential over high-entropy values. RealtimeTool takes an orthogonal approach by reducing the output representation itself while enabling parallel generation across independent streams; we provide a quantitative forward-pass comparison against jump-forward decoding in Appendix J.

**Token Compression and Pruning.**    A separate line of work reduces inference cost by shortening the token sequence the model must process or emit, e.g., prompt and KV-cache compression and token pruning that drop or merge low-information tokens. These methods operate on the input or on intermediate representations and are largely orthogonal to function-call output structure. RealtimeTool instead compresses the *output* representation of function calls by absorbing low-entropy structural and key tokens into special markers, and crucially repurposes those markers as mode selectors for parallel decoding—a property generic compression and pruning techniques do not provide.

**Efficient LLM Serving.**    Modern serving systems have improved LLM inference efficiency through various techniques. vLLM (Kwon et al., 2023) introduced PagedAttention for KV cache management, while SGLang (Zheng et al., 2024) proposed RadixAttention for prefix sharing. Quantization techniques such as AWQ (Lin et al., 2025) and GPTQ (Frantar et al., 2023) enable deployment on resource-constrained devices. RealtimeTool is compatible with these optimizations and can be combined with existing serving infrastructure.

**Real-Time LLM Applications.**    Vision-language-action models such as RT-2 (Zitkovich et al., 2023), OpenVLA (Kim et al., 2025), and $\pi_0$ (Black et al., 2025) have demonstrated promising capabilities for robotic control. In game AI, Voyager (Wang et al., 2024a) and SIMA (SIMA Team et al., 2024) explore open-world control capabilities. For GUI automation, AutoDroid (Wen et al., 2024) and Mobile-Agent (Wang et al., 2024b) explore LLM-powered device control. These application domains commonly face latency challenges that motivate our work on parallel decoding.

# B. Special Token Definitions

RealtimeTool introduces 17 special tokens organized into four functional groups. New tokens are appended to the vocabulary with IDs $|\mathcal{V}_{\text{base}}|$ through $|\mathcal{V}_{\text{base}}| + 16$. Embeddings are initialized by averaging semantically similar existing tokens.

**Example output format:**

```
<content>I'll check the weather for you.</content>
```

*Table 8.* Special token definitions and their functions.

| Group | Tokens | Count |
|---|---|---|
| Content Control | `<content>`, `</content>` | 2 |
| Function Control | `<function>`, `</function>` | 2 |
| Argument Control | `<arg`$k$`>`, `</arg`$k$`>` for $k \in \{1, \ldots, 6\}$ | 12 |
| Placeholder | `<|null|>` | 1 |
| **Total** | | **17** |

```
<function>get_weather</function>
<arg1>Beijing</arg1>
<arg2>2024-12-24</arg2>
<arg3>celsius</arg3>
<arg4><|null|></arg4>
<arg5><|null|></arg5>
<arg6><|null|></arg6>
```

## C. Parameter Count Statistics

We analyze the distribution of function parameter counts across all evaluation benchmarks to justify our design choice of 6 parallel argument heads.

### C.1. Distribution Analysis

Table 9 presents the number of samples with more than 6 parameters across all evaluation datasets (excluding SealTools 1–6 tools subsets, which are designed for controlled parameter count evaluation).

*Table 9.* Parameter count distribution across benchmarks. Samples with $>6$ parameters require sequential handling for additional arguments.

| Dataset | Total | >6 params | Ratio | Max Params |
|---|---|---|---|---|
| BFCL-v3 Exec Multiple | 50 | 0 | 0.00% | 6 |
| BFCL-v3 Exec Simple | 100 | 0 | 0.00% | 6 |
| BFCL-v3 Live Multiple | 1,053 | 167 | 15.86% | 21 |
| BFCL-v3 Live Simple | 258 | 14 | 5.43% | 10 |
| BFCL-v3 Multiple | 200 | 0 | 0.00% | 6 |
| BFCL-v3 Simple | 400 | 0 | 0.00% | 6 |
| SealTools (in-domain) | 199 | 4 | 2.01% | 8 |
| SealTools (out-domain) | 94 | 0 | 0.00% | 6 |
| Mobile Actions | 1,283 | 0 | 0.00% | 4 |
| OpenFunctions | 109 | 0 | 0.00% | 4 |
| ToolAlpaca (real) | 92 | 0 | 0.00% | 6 |
| ToolAlpaca (simulated) | 51 | 0 | 0.00% | 6 |
| **Total** | **3,889** | **185** | **4.76%** | – |

### C.2. Conclusion

Our default configuration of 6 parallel argument heads covers 95.2% of all function calls in the evaluated benchmarks. The 4.76% of cases with $>6$ parameters are concentrated in BFCL-v3 Live datasets, which contain real-world APIs with extensive optional parameters (e.g., REST APIs with many query parameters). For these cases, our model generates the first 6 arguments in parallel; additional arguments—typically optional—can be handled through sequential generation or omitted based on the tool schema's `required` field specification.

Notably, all synthetic and controlled benchmarks (BFCL Non-Live, SealTools, Mobile Actions, OpenFunctions, ToolAlpaca) have a maximum of 6 parameters, confirming that our head count aligns with common API design practices.

## D. Training Data Statistics

*Table 10.* Training data composition and statistics.

| Source | Samples | Percentage |
|---|---|---|
| xLAM-60K (filtered) | 48,000 | 40% |
| Synthetic (Qwen3-235B generated) | 70,000 | 58% |
| Special (pass-through functions) | 2,000 | 2% |
| **Total Groups** | **120,000** | **100%** |
| **Total Samples ($\times$8 heads)** | **960,000** | – |

**Argument distribution after augmentation:** 1 arg: 18%, 2 args: 20%, 3 args: 18%, 4 args: 17%, 5 args: 15%, 6 args: 12%. This balanced distribution addresses the natural imbalance where most real-world APIs use 1–2 arguments.

## E. Experimental Details

### E.1. Handling Parallel Tool Invocations

#### E.1.1. PROBLEM STATEMENT

Some benchmarks (e.g., Mobile Actions) contain samples with **parallel tool invocations**, where multiple functions are called simultaneously within a single turn. For example, a user request "Turn on my flashlight and show me the nearest bookstore on the map" expects two independent function calls:

```
tool_calls: [
  {"function": {"name": "turn_on_flashlight", "arguments": "{}"}},
  {"function": {"name": "show_map", "arguments": "{\"query\": \"nearest bookstore\"}"}}
]
```

Since RealtimeTool generates **one function call per inference pass**, we convert parallel invocations into sequential single-step evaluations using a history-based decomposition strategy.

#### E.1.2. CONVERSION STRATEGY

For $K$ parallel tool calls $\{c_1, c_2, ..., c_K\}$, we generate $K$ training/evaluation samples. Each sample $i$ receives:

- The original user query
- A `history` field containing preceding calls $\{c_1, ..., c_{i-1}\}$ (empty for the first entry)
- Ground truth expecting only call $c_i$

**Key design choice for training:** During training data preparation, we apply **random shuffling** to the prefix calls for each version. This teaches the model that parallel calls have no inherent order dependency, which aligns with our parallel decoding assumption.

#### E.1.3. EXAMPLE

**Original** (2 parallel calls: `turn_on_flashlight`, `show_map`):

**Training Version 1** (target: `show_map`):

```
User: "Turn on flashlight and show nearest bookstore"
History: [turn_on_flashlight()]
Assistant: -> show_map(query="nearest bookstore")   [TARGET]
```

**Training Version 2** (target: `turn_on_flashlight`):

```
User: "Turn on flashlight and show nearest bookstore"
History: [show_map(query="nearest bookstore")]  [SHUFFLED]
Assistant: -> turn_on_flashlight()  [TARGET]
```

### E.1.4. IMPLEMENTATION

Algorithm 1 illustrates the conversion process.

---

**Algorithm 1** Parallel Tool Call Decomposition

---

**Require:** Sample with query $q$, tool definitions $T$, parallel calls $C = \{c_1, ..., c_K\}$
**Ensure:** Evaluation entries $E = \{e_1, ..., e_K\}$
 1: $E \leftarrow \emptyset$
 2: **for** $i = 1$ to $K$ **do**
 3: $\quad prefix \leftarrow \{c_1, ..., c_{i-1}\}$
 4: $\quad prefix \leftarrow \text{Shuffle}(prefix)$ {Random order for training}
 5: $\quad history \leftarrow \text{Format}(prefix)$
 6: $\quad query_i \leftarrow \text{Concat}(environment, history, q)$
 7: $\quad gt_i \leftarrow \text{CreateGroundTruth}(c_i, T)$
 8: $\quad e_i \leftarrow (query_i, T, gt_i)$
 9: $\quad E \leftarrow E \cup \{e_i\}$
10: **end for**
11: **return** $E$

---

### E.1.5. BENEFITS

1. **Data augmentation**: $K$ parallel calls $\rightarrow K$ training samples
2. **Order invariance**: Random shuffling teaches the model that parallel calls have no causal dependencies
3. **Complete coverage**: Every function in the parallel set is learned as a prediction target
4. **Consistent evaluation**: Each decomposed entry is evaluated independently; a parallel invocation is considered fully correct only if all $K$ individual entries are correctly predicted

### E.2. Parameter Normalization

To ensure consistent evaluation across all benchmarks, we normalize tool definitions to have at most 6 parameters per function. Parameters are ordered as follows:

1. Required parameters in their original specification order
2. Optional parameters in alphabetical order

This normalization aligns with RealtimeTool's 6-argument head design and ensures that ground truth parameter ordering matches the tool definitions provided to the model.

### E.3. Latency Measurement Methodology

We measure inference latency using consistent methodology across all models and frameworks to ensure fair comparison.

**Measurement Protocol.**

- **Metric**: End-to-end wall-clock time from request submission to completion, including all framework overhead.
- **Warmup**: 5 requests discarded before measurement to ensure stable GPU state and cache initialization.
- **Sample size**: Full test split (1,283 samples for Mobile Actions) to capture latency distribution.
- **Statistics**: P50 (median) and P90 reported; P50 reflects typical performance while P90 captures tail latency.

**Framework Configuration.** All vLLM experiments use:

- enable_prefix_caching=True
- gpu_memory_utilization=0.85
- tensor_parallel_size=1
- Single request per batch (serial mode) to measure per-request latency rather than throughput

**Stop Token Configuration.** For RealtimeTool models, we configure both string-based and token-ID-based stopping to ensure immediate termination on special tokens:

```
stop = ["<|null|>", "</function>", "</arg1>", ..., "<|im_end|>"]
stop_token_ids = [tokenizer.encode(s)[-1] for s in stop]
```

This dual configuration is critical for achieving optimal latency, as token-ID-based stopping triggers immediately upon generation without the 1-step delay inherent in string matching.

### E.4. Mobile Actions Latency Benchmark Details

Table 11 provides comprehensive latency statistics for the Mobile Actions benchmark discussed in Section 3.4.

*Table 11.* Full latency statistics on Mobile Actions (1,283 samples, RTX 4090).

| Model | P50 (ms) | P90 (ms) | P95 (ms) | P99 (ms) | Mean (ms) | Avg Tok |
|---|---|---|---|---|---|---|
| Qwen3-4B (Baseline) | 468.5 | 667.0 | 731.2 | 845.3 | 489.7 | 39.2 |
| Qwen2.5-0.5B (Baseline) | 110.1 | 154.0 | 166.2 | 188.6 | 97.7 | 40.2 |
| FunctionGemma (270M) | 61.1 | 139.5 | 149.4 | 173.7 | 71.7 | 30.3 |
| **RT-Qwen-0.5B (Ours)** | **51.0** | **74.5** | **82.2** | **90.0** | **50.1** | $\sim$8 |

**Key Observations.**

- RT-Qwen-0.5B achieves the lowest latency across all percentiles.
- The P90/P50 ratio for RT-Qwen-0.5B (1.27) is significantly lower than FunctionGemma (2.27), indicating more consistent performance.
- Average output tokens for RT-Qwen-0.5B ($\sim$8) are 3–5$\times$ fewer than baselines (30–40), directly contributing to reduced decode time.

### E.5. Hardware Specifications

All experiments were conducted on the following hardware:

*Table 12.* Hardware configurations used in experiments.

| Component | Specification |
|---|---|
| GPU | NVIDIA RTX 4090 (24GB VRAM) |
| GPU | NVIDIA H100/H200 (80GB/141GB VRAM) |
| CUDA | 12.9 |
| PyTorch | 2.8.0 |
| vLLM | 0.12.0 |

Training experiments were conducted on H100/H200 GPUs. RT-Qwen-0.5B converges in approximately 3–4 H100-hours for initial training, and $\sim$1 GPU-hour for domain adaptation on Mobile Actions.

### E.6. Hierarchical KV Cache for Multi-Turn Deployment

For deployment in multi-turn, closed-loop control scenarios (e.g., robotic arms, mobile agents, game NPCs), we organize the shared prefix into a three-level KV cache hierarchy that maximizes reuse across both parallel heads and successive turns:

- **L1 (Session-persistent)**: System prompt and tool definitions. These remain fixed across an entire deployment session and are persistently cached. For a robotic control loop with a fixed tool schema, L1 is prefilled once at session start and reused for every subsequent control cycle.

- **L2 (Task-stable)**: Task-level instructions and environment descriptions that remain stable within a task episode but may change across episodes (e.g., a target object specification in a manipulation task). L2 sits on top of L1 and is invalidated only at episode boundaries.
- **L3 (Per-turn)**: Conversation history and per-turn environment observations, updated at every inference step.

Within a single inference step, all $H$ parallel heads share the combined L1+L2+L3 prefix—only the head-specific special token differs. Across turns, L1 and L2 are reused without recomputation, so prefill cost is amortized over the entire deployment session rather than paid per turn. In the limit (fixed tool schema, fixed task), only the L3 increment requires fresh prefill, reducing per-turn prefill to near-zero for long-running control loops. This hierarchical reuse is what makes the empirical latency reported in Sections 3.3–3.4 sustainable in continuous deployment, not just single-shot benchmarks.

## F. Training Hyperparameters

Table 13 presents the complete training configuration used for RealtimeTool models. We highlight several key design choices:

**LoRA Configuration.** We target MLP layers (gate_proj, up_proj, down_proj) rather than attention layers, as we find this provides greater capacity for learning the distinct output modes required by parallel heads. The relatively high rank ($r = 512$) is necessary to capture diverse head-specific patterns without interference (see ablation in Section 3.5).

**Head Loss Weighting.** We apply higher weights to the function head ($w_1 = 2.0$) since correct function selection is critical—an incorrect function name invalidates the entire output regardless of argument quality. Argument heads receive progressively increasing weights ($w_2$ through $w_7$) to counteract the natural data imbalance where higher-indexed arguments appear less frequently.

**Focal Weights.** For argument heads, we apply token-level focal weights that increase with head index. This addresses the challenge that later argument slots (arg4–arg6) have fewer training examples due to the skewed distribution of argument counts in real-world APIs.

## G. Batch Scaling Analysis

This appendix validates RealtimeTool's core assumption that parallel decoding incurs negligible overhead due to the memory-bandwidth-bound nature of autoregressive decoding.

### G.1. Experimental Setup

We measure per-token decode latency across varying batch sizes (1, 2, 4, 8, 16, 32, 64, 128) for six model configurations: Qwen2.5-0.5B, 1.5B, 3B, 7B, 14B, and Qwen3-4B. All experiments use identical input prompts and measure wall-clock time for the decode phase only, excluding prefill. We define **efficiency** as:

$$\text{Efficiency}(B) = \frac{T_d^{(B=1)}}{T_d^{(B)}} \tag{3}$$

where $T_d^{(B)}$ is the per-token decode latency at batch size $B$. An efficiency of 100% indicates perfect scaling (no overhead from batching); lower values indicate compute saturation.

### G.2. Results

Table 14 presents efficiency across batch sizes. The key finding is that **8-head parallel decoding (corresponding to B=8) achieves 93.0% average efficiency**, with per-token overhead of only +8.2% compared to single-sequence decoding.

Table 15 presents the same data as wall-clock overhead relative to B=1.

### G.3. Analysis

**Memory-Bound Regime.** The high efficiency at B=8 confirms that autoregressive decoding operates in a memory-bandwidth-bound regime for typical model sizes. The GPU's compute units remain underutilized during single-sequence

*Table 13.* Complete training hyperparameters for RealtimeTool in RT-Qwen3-4B

| Parameter | Value |
|---|---|
| *LoRA Configuration* | |
| Rank ($r$) | 512 |
| Alpha ($\alpha$) | 1024 |
| Dropout | 0.05 |
| Target Modules | gate_proj, up_proj, down_proj |
| *Learning Rates* | |
| Embedding | $1 \times 10^{-6}$ |
| LoRA | $1 \times 10^{-5}$ |
| LM Head | $1 \times 10^{-6}$ |
| Scheduler | Cosine with warmup |
| Warmup Ratio | 0.02 |
| *Training* | |
| Sequence Length | 2048 tokens |
| Batch Size (per device) | 8 groups |
| Gradient Accumulation | 4 steps |
| Effective Batch Size | 32 groups (256 samples) |
| Epochs | 3–4 |
| Precision | bfloat16 |
| Flash Attention | Enabled (v2) |
| *Head Loss Weights ($w_h$)* | |
| $w_0$ (content) | 0.01 |
| $w_1$ (function) | 2.0 |
| $w_{2..7}$ (arg1–arg6) | 1.1, 1.2, 1.3, 1.4, 1.5, 1.6 |
| *Token-Level Focal Weights ($\gamma$, non-null only)* | |
| $\gamma_{2,3}$ (arg1–arg2) | 1.0 |
| $\gamma_4$ (arg3) | 1.5 |
| $\gamma_5$ (arg4) | 2.0 |
| $\gamma_6$ (arg5) | 2.5 |
| $\gamma_7$ (arg6) | 3.0 |

decoding, allowing additional sequences to be processed with minimal latency increase.

**Efficiency Inflection Point.** Efficiency drops below 80% at B≥32 for most models, indicating the transition from memory-bound to compute-bound regime. RealtimeTool's 8 parallel heads remain well within the "comfort zone" where parallelization overhead is negligible.

**Model Size Effect.** Larger models (7B, 14B) maintain higher efficiency at larger batch sizes, as their higher memory bandwidth requirements keep the system memory-bound for longer. This suggests RealtimeTool's parallel decoding strategy scales favorably with model size.

**Implications for RealtimeTool.** With 8 parallel heads and +8.2% average overhead, the scheduling term $T_o(H)$ in Equation 2 is effectively absorbed into a small multiplicative factor on $T_d$, recovering the end-to-end latency of Equation 2:

$$T_{\text{ours}} \approx T_p + \max_i(N_i) \cdot T_d$$

Combined with the 4–6× token compression (Table 1), this yields the 3–6× end-to-end speedup observed in Section 3.3.

## H. Token Compression Details

This appendix provides detailed token compression statistics supporting the analysis in Section 2.1.

*Table 14.* Batch scaling efficiency (%) across model sizes. Higher values indicate better utilization of idle compute capacity. RealtimeTool uses 8 heads, corresponding to B=8.

| Model | B=1 | B=2 | B=4 | B=8 | B=16 | B=32 | B=64 | B=128 |
|---|---|---|---|---|---|---|---|---|
| Qwen2.5-0.5B | 100.0 | 96.4 | 95.6 | 90.6 | 81.4 | 67.3 | 52.8 | 37.1 |
| Qwen2.5-1.5B | 100.0 | 98.3 | 97.5 | 92.7 | 86.5 | 78.7 | 64.7 | 47.1 |
| Qwen2.5-3B | 100.0 | 99.1 | 98.4 | 93.5 | 89.8 | 83.5 | 72.1 | 56.2 |
| Qwen3-4B | 100.0 | 98.7 | 96.3 | 92.5 | 83.7 | 74.4 | 58.1 | 45.8 |
| Qwen2.5-7B | 100.0 | 87.2 | 91.5 | 91.5 | 90.9 | 84.4 | 75.0 | 62.0 |
| Qwen2.5-14B | 100.0 | 99.3 | 98.4 | 97.4 | 95.2 | 90.6 | 78.1 | 66.5 |
| **Average** | 100.0 | 96.5 | 96.3 | **93.0** | 87.9 | 79.8 | 66.8 | 52.5 |

*Table 15.* Wall-clock overhead (%) relative to single-sequence decoding (B=1). Lower values indicate less overhead from parallel execution.

| Model | B=1 | B=2 | B=4 | B=8 | B=16 | B=32 | B=64 | B=128 |
|---|---|---|---|---|---|---|---|---|
| Qwen2.5-0.5B | +0.0 | +3.7 | +4.6 | +10.3 | +19.6 | +34.5 | +62.7 | +125.3 |
| Qwen2.5-1.5B | +0.0 | +1.7 | +2.6 | +7.6 | +15.6 | +23.9 | +44.4 | +93.2 |
| Qwen2.5-3B | +0.0 | +0.9 | +1.6 | +6.9 | +10.0 | +19.1 | +37.0 | +76.9 |
| Qwen3-4B | +0.0 | +1.3 | +3.9 | +5.5 | +10.4 | +23.2 | +37.6 | +68.0 |
| Qwen2.5-7B | +0.0 | +12.8 | +15.7 | +16.3 | +21.6 | +26.6 | +41.1 | +68.7 |
| Qwen2.5-14B | +0.0 | +0.8 | +1.7 | +2.7 | +4.7 | +10.1 | +23.2 | +44.9 |
| **Average** | +0.0 | +3.5 | +5.0 | **+8.2** | +13.7 | +22.9 | +41.0 | +79.5 |

### H.1. Methodology

We measure token compression by comparing output lengths between baseline models (generating full JSON-formatted function calls) and RealtimeTool models (generating only value tokens across parallel heads). For RealtimeTool, we report the **bottleneck head** token count—the maximum across all parallel heads—as this determines end-to-end latency per Equation 2.

Statistics are computed exclusively on samples where **both** baseline and RealtimeTool models produce correct outputs, ensuring fair comparison on equivalent task difficulty. The compression ratio (CR) is defined as:

$$CR = \frac{\text{Baseline tokens}}{\text{RT bottleneck-head tokens}} \quad (4)$$

### H.2. Overall Results

Table 16 presents detailed statistics across model sizes, including mean, median (P50), and 90th percentile (P90) for both baseline and RealtimeTool token counts.

Key observations:

- **Consistent compression across scales**: All models achieve 4.3–5.4× mean compression, indicating that the structural redundancy in function call outputs is model-agnostic.
- **Higher median than mean**: P50 compression (4.7–6.0×) exceeds mean compression, suggesting that outliers with longer outputs (complex nested arguments) pull down the average.
- **Stable RT token counts**: RealtimeTool bottleneck-head tokens remain remarkably stable (8.7–9.0 mean) across model sizes, reflecting the consistent information content of function calls regardless of baseline verbosity.

### H.3. Per-Benchmark Breakdown

Table 17 presents compression ratios by benchmark group, revealing how compression varies across task types.

Key observations:

- **Highest compression on simple APIs**: BFCL Non-Live and OpenFunc achieve 5.7–7.9× compression, as these benchmarks feature functions with fewer, simpler arguments.

*Table 16.* Detailed token compression analysis. BL = Baseline tokens, RT = RT bottleneck-head tokens, CR = Compression Ratio. All statistics computed on samples where both Baseline and RT models produce correct outputs.

| | Mean | | | P50 (Median) | | | P90 | | | |
|---|---|---|---|---|---|---|---|---|---|---|
| **Model** | BL | RT | CR | BL | RT | CR | BL | RT | CR | **N** |
| Qwen2.5-0.5B | 44.1 | 8.7 | 5.06× | 36.0 | 6.0 | 6.00× | 69.0 | 17.0 | 4.06× | 3328 |
| Qwen2.5-1.5B | 38.3 | 8.9 | 4.32× | 33.0 | 7.0 | 4.71× | 64.0 | 20.0 | 3.20× | 3590 |
| Qwen2.5-3B | 39.2 | 8.9 | 4.38× | 33.0 | 7.0 | 4.71× | 64.0 | 20.0 | 3.20× | 3939 |
| Qwen3-4B† | 38.4 | 8.9 | 4.32× | 34.0 | 7.0 | 4.86× | 65.0 | 19.0 | 3.42× | 4074 |
| Qwen2.5-7B | 48.3 | 9.0 | 5.35× | 35.0 | 7.0 | 5.00× | 68.0 | 20.0 | 3.40× | 4094 |
| Qwen2.5-14B | 40.2 | 8.9 | 4.51× | 34.0 | 7.0 | 4.86× | 66.0 | 19.0 | 3.47× | 3946 |

†: Qwen3 backbone. N = number of both-correct samples.

*Table 17.* Token compression ratio by benchmark group. CR = Compression Ratio (Baseline / RT). Results on both-correct samples. BFCL-NL = BFCL Non-Live, BFCL-L = BFCL Live, BFCL-E = BFCL Exec.

| **Model** | **BFCL-NL** | **BFCL-L** | **BFCL-E** | **SealTools** | **MobileAct** | **OpenFunc** | **ToolAlpaca** | **Overall** |
|---|---|---|---|---|---|---|---|---|
| Qwen2.5-0.5B | 7.05× | 5.52× | 4.53× | 5.16× | 4.17× | 7.91× | 5.14× | 5.06× |
| Qwen2.5-1.5B | 5.67× | 5.00× | 3.85× | 4.51× | 3.38× | 5.19× | 6.04× | 4.32× |
| Qwen2.5-3B | 5.73× | 4.83× | 3.55× | 4.53× | 3.73× | 5.20× | 4.48× | 4.38× |
| Qwen3-4B† | 5.71× | 4.89× | 3.16× | 4.55× | 3.54× | 5.17× | 4.47× | 4.32× |
| Qwen2.5-7B | 6.77× | 6.43× | 3.91× | 5.97× | 3.46× | 5.29× | 20.54× | 5.35× |
| Qwen2.5-14B | 5.81× | 5.44× | 3.38× | 4.91× | 3.31× | 5.17× | 6.16× | 4.51× |

†: Qwen3 backbone.

- **Lower compression on execution tasks**: BFCL Exec and MobileAct show 3.2–4.5× compression, reflecting more complex argument structures (e.g., nested objects, longer string values).
- **Outlier in ToolAlpaca**: Qwen2.5-7B achieves 20.54× compression on ToolAlpaca, likely due to baseline verbosity on specific samples; this outlier does not significantly affect overall conclusions.

### H.4. Relationship to End-to-End Speedup

The token compression ratio provides a **theoretical upper bound** on decode-phase speedup. The actual end-to-end speedup (3–6×, Section 3.3) is lower due to:

1. **Prefill overhead**: $T_p$ is identical for both methods and does not benefit from token compression.
2. **Parallelization overhead**: The +8.2% overhead from 8-head parallel decoding (Appendix G).
3. **Framework overhead**: Scheduling, memory allocation, and other system-level costs in inference frameworks.

Despite these factors, RealtimeTool achieves 70–80% of the theoretical compression-based speedup in practice, demonstrating efficient translation of token reduction into latency improvement.

## I. Limitations: Parameter Independence Assumption

### I.1. The Independence Assumption

Our parallel decoding assumes that function arguments are **conditionally independent** given the user query and function name:

$$P(\arg_1, ..., \arg_k | q, f) \approx \prod_{i=1}^{k} P(\arg_i | q, f) \tag{5}$$

This assumption is justified by the design philosophy of JSON-based function calling: parameters are defined as unordered key-value pairs, and well-designed APIs treat each parameter as capturing an independent aspect of the request.

### I.2. Empirical Validation

Our benchmark results provide strong empirical support for this assumption:

- **Accuracy improvement**: RealtimeTool achieves equal or higher accuracy than sequential baselines across all benchmarks (Table 2). If parameter dependencies were prevalent and important, removing sequential conditioning should *decrease* accuracy—the opposite of what we observe.
- **Order invariance in training**: Our parallel-to-multi-turn conversion (Appendix E.1) shuffles parameter order during training. The model's strong performance indicates it learns to predict parameters independently of their presentation order.

### I.3. Theoretical Edge Cases

We acknowledge that some API designs could introduce parameter dependencies:

**Example 1: Runtime-dependent validation**

```
read_file_range(file_path, start_line, end_line)
```

Here, `end_line`'s validity depends on the actual file length. However, this is a *runtime* constraint, not a *generation* dependency—a sequential model would also generate `end_line=100` if the user requests it, regardless of whether the file has 100 lines.

**Example 2: Cascading inference**

```
format_convert(input_string, detected_format, output_format)
```

If `detected_format` must be inferred from `input_string`, this creates a true dependency. However:

1. This is poor API design—`detected_format` should be computed internally by the function, not passed as a parameter.
2. In practice, both values can often be inferred independently from context (e.g., "convert 2024-01-15 to MM/DD/YYYY" allows inferring both the input format and the input value).

### I.4. Handling True Dependencies

For the rare cases where genuine parameter dependencies exist, RealtimeTool supports two fallback strategies:

1. **Consolidation**: Dependent parameters can be combined into a single head as a structured value (e.g., a list or nested object).
2. **Multi-turn refinement**: For complex cases, the model can generate a partial call first, then refine in subsequent turns based on execution feedback.

### I.5. Conclusion

The parameter independence assumption is both theoretically grounded (JSON's key-value design) and empirically validated (accuracy improvements). While edge cases exist, they represent either poor API design or runtime constraints that affect sequential models equally. Our 95.2% benchmark coverage with 6 heads (Appendix C) further confirms that real-world APIs align with this assumption.

## J. Comparison with Constrained Decoding

As discussed in Section 1, constrained-decoding methods fall into two categories with very different latency implications. Logits-masking frameworks (Outlines (Willard & Louf, 2023), XGrammar (Dong et al., 2024)) restrict the output to a valid grammar but still issue one forward pass per token, and the masking step adds per-token overhead; they therefore do not reduce end-to-end generation latency. The technique that does reduce forward passes is jump-forward decoding (as in SGLang (Zheng et al., 2024)), which emits structurally deterministic spans (delimiters, fixed keys) without a model call.

**Why we report a theoretical upper bound.** Jump-forward decoding has been effectively deprecated in current SGLang releases, which makes a faithful end-to-end wall-clock benchmark on the current stack unreliable. We therefore compare against its *theoretical optimum*: we assume jump-forward removes *every* structurally deterministic token in the baseline JSON output (an idealization that no real implementation attains, since it ignores masking and scheduling overhead), and we count the remaining forward passes. This gives jump-forward the most favorable possible treatment.

*Table 18.* Forward-pass comparison on BFCL-v3 (Qwen3-4B). Jump-forward is given its theoretical upper bound (all deterministic tokens removed, no overhead). RealtimeTool counts the bottleneck-head tokens that determine end-to-end latency.

| Method | Avg. Fwd Passes | Speedup |
|---|---|---|
| Vanilla (unconstrained) | 40.7 | 1.00× |
| Jump-forward (upper bound) | 29.5 | 1.38× |
| RealtimeTool (bottleneck head) | **9.4** | **4.32×** |

As shown in Table 18, even at its theoretical optimum jump-forward attains only $1.38\times$, versus $4.32\times$ for RealtimeTool—a gap of more than $3\times$. The reason is structural: jump-forward can only skip predictable *structural* tokens, leaving every high-entropy value to be decoded sequentially, whereas RealtimeTool both compresses those structural tokens and decodes the remaining values as parallel streams. In practice the realized jump-forward speedup would be strictly lower than this upper bound due to masking and scheduling overhead.

# K. Capability Preservation and Tool-Space Scaling

### K.1. General Capability Is Preserved

A natural concern is whether specializing a model for parallel function-call decoding erodes its general-purpose ability. It does not. RealtimeTool's multi-head modes are activated *only* by the special tokens appended to the prompt; under standard prompting the model follows its ordinary autoregressive path. We evaluate RT-Qwen3-4B and its baseline under identical default prompting (no special tokens triggered) on two general benchmarks (Table 19). RT-Qwen3-4B retains essentially identical MMLU accuracy and slightly improves on instruction following, indicating that the LoRA adaptation adds a fast path for function calling without overwriting pretrained knowledge.

*Table 19.* General-capability comparison under standard prompting (no special tokens). The parallel-decoding adaptation does not degrade general ability.

| Benchmark | Qwen3-4B | RT-Qwen3-4B | Δ |
|---|---|---|---|
| MMLU (5-shot) | 72.64 | 72.35 | −0.29 |
| IFEval (prompt-level) | 84.10 | 86.88 | +2.78 |
| IFEval (instruction-level) | 89.45 | 91.37 | +1.92 |

### K.2. Scaling with the Number of Candidate Tools

We further analyze how accuracy varies with the number of candidate tools presented in the prompt, using a stratified breakdown of Qwen3-4B over the full evaluation suite (Table 20). RealtimeTool maintains stable function accuracy (97.8–99.9%) from 1 to 7 candidate tools with no observable degradation, whereas the baseline drops sharply at 7 tools (function accuracy $93\% \rightarrow 75.3\%$). We attribute the baseline's degradation to its substantially longer per-call output, which increases the chance of a generation error—an effect that is most pronounced at the small model scales (0.5B–4B) targeted by this work. Strata with $\geq 9$ tools contain too few samples for statistical conclusions and are included only for completeness. When the candidate set grows to hundreds of tools, the bottleneck shifts to the prompt side—fitting definitions in context and discriminating similar tools—which affects sequential and parallel decoding equally; retrieval-based pre-filtering, orthogonal to our method, is the standard remedy.

# L. On-Device Deployment

To validate that RealtimeTool's server-side speedups translate to genuine edge hardware, we deploy RT-Qwen-0.5B and FunctionGemma on a smartphone and compare them under an apples-to-apples configuration. Both models run through the

*Table 20.* Accuracy stratified by number of candidate tools (Qwen3-4B, full evaluation suite, 5,294 samples). Overall / Function accuracy (%).

| # Tools | Samples | RealtimeTool | | Baseline | |
|---|---|---|---|---|---|
| | | Overall | Func | Overall | Func |
| 1 | 1,145 | 85.6 | 99.9 | 82.3 | 94.8 |
| 2 | 563 | 84.0 | 98.8 | 82.2 | 94.7 |
| 3 | 688 | 84.5 | 97.8 | 82.4 | 93.0 |
| 4 | 488 | 85.3 | 98.0 | 85.3 | 93.2 |
| 5 | 426 | 85.2 | 97.9 | 80.8 | 92.0 |
| 6 | 650 | 89.5 | 99.7 | 85.5 | 93.1 |
| 7 | 1,310 | 84.4 | 99.9 | 68.7 | 75.3 |
| *Sparse strata (too few samples for conclusions)* | | | | | |
| 9 | 11 | 81.8 | 90.9 | 90.9 | 100.0 |
| 10 | 12 | 100.0 | 100.0 | 100.0 | 100.0 |
| 26 | 1 | 100.0 | 100.0 | 100.0 | 100.0 |

same llama.cpp binary with INT8 (Q8_0) weights on an iPhone 17 Pro Max (A19 Pro), evaluated on the Mobile Actions test split (961 samples). Table 21 reports end-to-end latency and sustained thermal state.

*Table 21.* On-device comparison on iPhone 17 Pro Max (A19 Pro), llama.cpp, INT8 (Q8_0), Mobile Actions (961 samples). Lower latency and lower thermal pressure are better.

| Model | Params | P50 | P90 | Thermal state |
|---|---|---|---|---|
| FunctionGemma | 270M | 709 ms | 1570 ms | 78.7% fair / 21.3% serious |
| RT-Qwen-0.5B | 500M | **528 ms** | **1018 ms** | 64.7% nominal / 35.3% fair |

Despite having roughly $1.8\times$ more parameters, RT-Qwen-0.5B is $1.34\times$ faster at P50 and $1.54\times$ faster at P90, and sustains a substantially cooler thermal state—fewer decoding steps directly reduce sustained compute and thus thermal load, which governs long-running on-device throughput. These numbers are conservative: llama.cpp provides neither prefix caching nor an optimized batching backend (unlike vLLM/SGLang), both of which would further benefit RealtimeTool's shared-prefix parallel decoding. This experiment substantiates the deployability claims discussed in Section 3.4.

## M. Compatibility with Speculative Decoding

As discussed in Section 1, speculative decoding methods are orthogonal to RealtimeTool: they accelerate per-token generation through draft-and-verify mechanisms, while we reduce the total number of tokens to generate. This appendix validates that these approaches can be combined for additional benefits.

### M.1. Experimental Setup

We evaluate speculative decoding on RealtimeTool-trained models using the standard draft-model approach (Leviathan et al., 2023). The target model is RT-Qwen2.5-14B, with RT-Qwen2.5-0.5B and RT-Qwen2.5-1.5B serving as draft models. All models share the same special token vocabulary and output format, enabling direct speculation without format conversion.

We test speculation depths $N \in \{2, 3, 4\}$ (number of tokens drafted per step) across all BFCL-v3 subsets and report forward pass reduction (vanilla forwards / speculative forwards) and token acceptance rate (accepted tokens / drafted tokens).

### M.2. Results

Table 22 summarizes the results across draft model sizes and speculation depths.

Table 23 provides per-dataset breakdown for the best-performing configuration (RT-Qwen-1.5B draft, $N = 4$).

*Table 22.* Speculative decoding results with RT-Qwen2.5-14B as target. Speedup = forward pass reduction (higher is better). Accept Rate = proportion of drafted tokens accepted by target model.

| Draft Model | N | Avg Speedup | Avg Accept Rate |
|---|---|---|---|
| RT-Qwen-0.5B | 2 | 2.35× | 95.0% |
| | 3 | 2.83× | 94.0% |
| | 4 | 3.22× | 93.6% |
| RT-Qwen-1.5B | 2 | 2.37× | 96.1% |
| | 3 | 2.87× | 95.5% |
| | 4 | 3.24× | 95.1% |

*Table 23.* Per-dataset speculative decoding results (RT-Qwen-1.5B draft, $N = 4$).

| Dataset | Samples | Vanilla Fwds | Spec Fwds | Speedup | Accept Rate |
|---|---|---|---|---|---|
| BFCL-v3 Simple | 400 | 6,578 | 2,031 | 3.24× | 95.3% |
| BFCL-v3 Live Simple | 244 | 4,698 | 1,401 | 3.35× | 93.0% |
| BFCL-v3 Multiple | 200 | 3,333 | 1,096 | 3.04× | 96.1% |
| BFCL-v3 Live Multiple | 886 | 17,196 | 6,593 | 2.61× | 90.0% |
| BFCL-v3 Exec Simple | 100 | 1,844 | 489 | 3.77× | 98.2% |
| BFCL-v3 Exec Multiple | 50 | 986 | 285 | 3.46× | 98.1% |
| **Total/Average** | 1,880 | 34,635 | 11,895 | **3.24×** | **95.1%** |

## M.3. Analysis

**High acceptance rates.** Both draft models achieve >93% token acceptance rates across all configurations, significantly higher than typical speculative decoding scenarios on general text generation (70–85%). This is because RealtimeTool's simplified output format—consisting primarily of function names and argument values—exhibits lower entropy and higher predictability than free-form text, making smaller models more effective as drafters.

**Consistent speedup across datasets.** Forward pass reduction ranges from 2.6× to 3.8× depending on dataset complexity. Simpler datasets (Exec Simple/Multiple) achieve higher speedup due to more predictable outputs, while Live Multiple shows lower speedup due to greater output diversity.

**Draft model size trade-off.** The 1.5B draft model achieves slightly higher acceptance rates (+1–2%) than the 0.5B model, translating to marginally better speedup. However, the 0.5B model may be preferable for memory-constrained deployments where loading a larger draft model is impractical.

**Speculation depth trade-off.** Increasing $N$ from 2 to 4 improves speedup from ~2.4× to ~3.2×, with only minor degradation in acceptance rate (~1%). This suggests $N = 4$ as a practical default for RealtimeTool models.

## M.4. Combined Speedup Potential

The total theoretical speedup when combining RealtimeTool with speculative decoding is multiplicative:

$$\text{Speedup}_{\text{combined}} = \text{CR} \times \text{Speedup}_{\text{spec}} \tag{6}$$

where CR is the token compression ratio from RealtimeTool (4–6×, Table 1) and $\text{Speedup}_{\text{spec}}$ is the forward pass reduction from speculative decoding (2–3×).

For RT-Qwen2.5-14B with CR $\approx 4.5\times$ (Table 1) and 1.5B draft model at $N = 4$ (3.24× forward reduction), the combined theoretical speedup reaches approximately **14.6×** in forward passes compared to vanilla autoregressive decoding of the baseline 14B model.

**Practical considerations.** While forward pass reduction provides a useful proxy for latency improvement, actual wall-clock speedup depends on additional factors: draft model inference overhead, verification batch size, and memory bandwidth utilization. For memory-bound inference on consumer GPUs, RealtimeTool's token compression alone often saturates available bandwidth, making speculative decoding most beneficial for larger models or datacenter deployments where compute becomes the bottleneck. We leave detailed wall-clock benchmarking of combined approaches to future work.

## M.5. Conclusion

These results confirm that RealtimeTool and speculative decoding are complementary acceleration strategies operating on different axes: RealtimeTool reduces *what* to generate (fewer tokens through compression and parallelization), while speculative decoding accelerates *how* to generate (fewer sequential forward passes through draft-and-verify). The exceptionally high acceptance rates ($>93\%$) suggest that RealtimeTool's simplified output format is particularly amenable to speculative approaches, opening promising directions for further optimization.

