# OpenReview forum: "RealtimeTool: Parallel Decoding for Real-Time LLM Function Calling"
_ICML.cc/2026/Conference — ICML 2026 regular_

### Official Review · Reviewer_UYJK · 2026-03-04

**Soundness:** 2
**Presentation:** 3
**Significance:** 2
**Originality:** 2
**Overall Recommendation:** 3
**Confidence:** 2

**Summary:**

This paper targets accelerating inference for LLM function-calling on edge devices. The core idea is clear and intuitive: because many tokens in function-calling outputs have low inter-dependency, they can be predicted directly from the input query rather than autoregressively conditioning on previously generated tokens, enabling parallel decoding. This introduces an explicit trade-off between generality and speed: by specializing the model/decoding for function calling, the method achieves substantial inference speedups.

**Compliance With Llm Reviewing Policy:**

Affirmed.

**Final Justification:**

Thanks for the authors' detailed response. I still think this method sacrifices generalization for efficiency, which makes it not a scalable solution. However, as I'm not familiar with edge device deployment, I'm willing to decrease my confidence score from 4 to 2.

**Key Questions For Authors:**

Please see weakness

**Limitations:**

yes

**Strengths And Weaknesses:**

## Strengths
1. **Clear and reasonable motivation/idea.** The independence assumption for function-calling fields is plausible and the parallel-decoding direction is easy to understand.
2. **Comprehensive experiments.** The evaluation covers a wide range of model sizes (0.5B–14B) and considers integration with practical inference techniques such as quantization.
3. **Well written.** The paper is easy to follow and communicates the method effectively.

## Weaknesses
1. **Limited generality.** The approach appears tailored to function-calling outputs and does not obviously extend to broader generation tasks.
2. **Generality–capability trade-off.** LLMs are valued as general-purpose generative models, but this method effectively narrows them toward a task-specific discriminative model. This complicates or limits settings that rely on standard autoregressive generation, e.g., multi-turn generations (where outputs depend on conversational context and prior turns) or workflows that generate intermediate reasoning/thoughts before producing the final function call.
3. **Limited benefit when generated with reasoning**. Intermediate thinking/reasoning steps have been shown to improve performance significantly. If a model must generate such intermediate reasoning before producing the final function call, the proposed parallel decoding would apply to only a small portion of the output, substantially limiting its practical speedup.

---

> ### Author Rebuttal · Authors · 2026-03-30
>
> We sincerely thank you for your clear and insightful evaluation. We greatly appreciate your recognition that our motivation is "clear and reasonable," experiments are "comprehensive," and the paper is "well written." Your concern regarding the generality–capability trade-off is perceptive, and we address each weakness with new evidence below.
>
> ## W1. Limited Generality & W2. Generality–Capability Trade-off
>
> RealtimeTool is not a trade-off that sacrifices general capability, but a specialization that preserves it. Unlike VLA models (RT-2, OpenVLA, π0) that fully repurpose VLMs for action generation, RealtimeTool adds a parallel decoding fast-path while keeping the base model's general capabilities intact, as demonstrated below.
>
> **Capability preservation.** We evaluate RT-Qwen3-4B and Baseline model on two general-purpose benchmarks under identical default prompting(no special tokens triggered):
>
> | Benchmark | Qwen3-4B (Baseline) | RT-Qwen3-4B | Δ |
> |---|---|---|---|
> | MMLU (5-shot) | 72.64% | 72.35% | −0.29% |
> | IFEval (prompt-level) | 84.10% | 86.88% | +2.78% |
> | IFEval (instruction-level) | 89.45% | 91.37% | +1.92% |
>
> RT-Qwen3-4B retains virtually identical MMLU performance and improves on instruction following (+2.78%). This is because RealtimeTool's multi-head modes are activated exclusively by special tokens appended to the prompt—under standard prompting, the model behaves identically to the baseline. The LoRA adaptation does not degrade general capabilities.
>
> **Content head and multi-turn in action.** Among our 8 heads, we reserve a content head (`<content>`) for natural language output alongside function calls (Appendix B). Combined with our history-based multi-turn decomposition (Appendix E.1, Algorithm 1), this enables scenarios like a mobile assistant that simultaneously acts and responds.
>
> *Example 1 (sampled directly from the Mobile Actions benchmark):*
>
> User: *"Please send an email to javier.ortega@ecotradeintl.com with subject 'Update ..."*
>
> | Head | Output |
> |---|---|
> | content | **Sending the email now.** |
> | function | send\_email |
> | arg1 | javier.ortega@ecotradeintl.com |
> | arg2 | Update on Q4 Report |
> | arg3 | I've uploaded the revised figures... |
> | arg4–6 | ∅ (null) |
>
> *Example 2 (manually constructed multi-turn follow-up on Example 1):*
>
> History contains Example 1's call: `send_email(to="javier.ortega@...", subject="Update on Q4 Report", body="...")`. User: *"Could you pull up the Ecotradeintl headquarters in Barcelona on the map?"*
>
> | Head | Output |
> |---|---|
> | content | **Pulling up the location for you now.** |
> | function | show\_map |
> | arg1 | Ecotradeintl headquarters, Barcelona |
> | arg2–6 | ∅ (null) |
>
> The content head produces natural conversational responses while function heads generate actions in parallel—all sharing the same prefix KV cache via our three-level hierarchy (Appendix E.1.4):
>
> **L1** system prompts + tool definitions, persistently cached across turns;
> **L2** task-level instructions, stable within a session—not used in this conversational example, but essential for closed-loop control scenarios such as robotic arm control (see W3 below);
> **L3** history and environment, updated per turn. This hierarchy maximizes KV cache reuse. The model retains conversational fluency, multi-turn context tracking, and function calling accuracy simultaneously.
>
> ## W3. Limited Benefit with Reasoning
>
> We acknowledge that when a model generates extended reasoning chains before a function call, RealtimeTool accelerates only the action portion. However, our target applications *inherently cannot use CoT/ReAct*:
>
> - **Real-time control (10 Hz):** A robotic arm or game AI agent has ~100ms per control cycle. There is no time budget for reasoning tokens—the function call is the entire output. We provide a demo of a robotic arm in NVIDIA Isaac Sim ([https://anonymous.4open.science/r/temp_icml-C301/](https://anonymous.4open.science/r/temp_icml-C301/), Fig. 1: Franka robot arm powered by RealtimeTool).
> - **Edge deployment:** On-device models (0.5B–4B) lack the capacity for effective chain-of-thought; they are deployed as direct action generators.
>
> This is not a niche scenario—it represents a rapidly growing deployment paradigm:
>
> - **Apple Foundation Models (WWDC 2025):** Apple's on-device ~3B model framework lists constrained decoding for tool calling as a key capability, supporting low-latency parallel/serial tool call scheduling on-device.
> - **NVIDIA ACE (CES 2025):** NVIDIA deploys edge LLM for game AI, using function calling to achieve real-time AI NPC control at human-like response frequencies.
> - **Google FunctionGemma (Dec 2025):** Google specifically distilled Gemma to 270M parameters for edge function calling—our RT-Qwen-0.5B outperforms it in both accuracy and latency (Tables 4–5), with further validation on iPhone deployment (detailed in response to Reviewer LAba, Q1).
>
> We sincerely thank you again for your valuable evaluation and constructive suggestions!

---

> > ### Author Rebuttal · Reviewer_UYJK · 2026-04-02
> >
> > I will maintain my score, as the proposed method does not appear to be well aligned with current LLM paradigms such as Chain-of-Thought (CoT) and ReAct.
> >
> > That said, since I am not very familiar with edge devices and robotic control, I am willing to lower my confidence.
> >
> > From my limited understanding, hardware is becoming increasingly powerful, and small models are also showing strong capabilities. For example, 4B-scale models can already achieve over 80% on AIME. Therefore, incompatibility with CoT/ReAct could be a significant limitation for future applications. In addition, there are more general methods, such as multi-token prediction (MTP) and speculative decoding, that may offer comparable efficiency benefits.

---

> > > ### Author Response · Authors · 2026-04-02
> > >
> > > Thank you for acknowledging that our concerns have been resolved, and for your note about confidence. We genuinely appreciate your continued engagement. We'd like to briefly discuss the new points you raised.
> > >
> > > ## CoT/ReAct and Our Positioning
> > >
> > > RealtimeTool and CoT/ReAct address orthogonal problem spaces. CoT/ReAct improve reasoning quality for complex, multi-step tasks where latency requirements are relatively relaxed; RealtimeTool reduces end-to-end generation latency for scenarios where the function call itself is the entire output and must complete within a tight time budget.
> > >
> > > In these latency-critical settings — such as the robotic control, game AI, and mobile agent scenarios discussed in our paper and previous rebuttals — CoT/ReAct are not merely suboptimal; they are physically infeasible. There is no time budget for sequential reasoning tokens within a single control cycle. This is a well-recognized challenge: the Efficient VLA Survey (arxiv 2510.17111) identifies autoregressive sequential dependency as a bottleneck that "hampers real-time embodied control," and current VLA models like OpenVLA (7B) reach only ~6 Hz while real-time tasks typically require 50–100 Hz (KV-Efficient VLA, arxiv 2509.21354).
> > >
> > > This gap stems from autoregressive decoding itself, and hardware improvements alone cannot close it — faster hardware benefits all methods equally but does not change the sequential nature of token generation. What hardware progress does enable is using RealtimeTool with stronger models. You mentioned that small models like Qwen3-4B are showing impressive capabilities — as you noted, 4B-scale models already achieve over 80% on AIME — and we fully agree. This rapid improvement in model quality through better architectures, training, and data is exactly why the speed bottleneck matters now more than ever. RealtimeTool complements this progress by solving the speed side: it allows these increasingly capable small models to be deployed in scenarios where autoregressive latency previously made LLM-based approaches infeasible, as demonstrated in our Isaac Sim and on-device iPhone deployments presented in the previous rebuttal. Today we achieve 16 Hz real-time control with Qwen3-4B (Table 3); with future hardware, the same method could enable 32 Hz at 4B scale, or 16 Hz with 30B-class models — each combination unlocking new application tiers.
> > >
> > > ## Speculative Decoding and MTP
> > >
> > > You suggested that speculative decoding and MTP might offer comparable benefits. These are valuable general-purpose acceleration techniques, but they face a fundamental ceiling in short-output scenarios.
> > >
> > > Leviathan et al. (2023) state after Theorem 3.8 (Section 3.3):
> > >
> > > > "Note that Theorem 3.8 assumes long enough generations (for example, since we run M_p at least once, the improvement factor is capped by the number of generated tokens)."
> > >
> > > For function calls with only 8–40 output tokens, this cap is severe. The fixed overhead of each draft-verify cycle — loading the draft model, running verification forward passes — occupies a much larger proportion of total inference time compared to long-generation tasks like chain-of-thought, where this overhead is amortized over hundreds of tokens. As a result, the practical speedup in short-output scenarios is far below what these methods achieve in their primary use cases. MTP (Gloeckle et al., 2024) integrates speculation into training, but the same output-length ceiling and overhead amplification apply at inference.
> > >
> > > Beyond this theoretical cap, both methods operate along the sequential token chain and cannot exploit the structural independence between function arguments — which is the core property RealtimeTool leverages. We discuss this orthogonality in Section 1.2 of our paper, and Appendix J validates it empirically: combining speculative decoding with RealtimeTool yields >93% acceptance rate and up to 3.24× additional forward pass reduction, confirming that they complement rather than substitute each other.
> > >
> > > ## Closing
> > >
> > > We recognize that our work sits outside mainstream LLM paradigms, and we appreciate your perspective on this. We believe this positioning reflects the nature of the problem: existing paradigms and acceleration methods are not designed for — and face fundamental limits in — real-time control scenarios. We hope these discussions help address your remaining questions, and we are happy to discuss further.
> > >
> > > **References:**
> > > - Efficient VLA Survey: https://arxiv.org/abs/2510.17111
> > > - Yu et al. (2025), Survey on Efficient VLAs: https://arxiv.org/abs/2510.24795
> > > - KV-Efficient VLA: https://arxiv.org/abs/2509.21354
> > > - Leviathan et al. (2023), Speculative Decoding: https://arxiv.org/abs/2211.17192
> > > - Gloeckle et al. (2024), Multi-Token Prediction: https://arxiv.org/abs/2404.19737
> > > - Our paper: Section 1.2, Appendix J, Table 3

---

### Official Review · Reviewer_LAba · 2026-03-11

**Soundness:** 3
**Presentation:** 3
**Significance:** 3
**Originality:** 3
**Overall Recommendation:** 4
**Confidence:** 3

**Summary:**

This paper proposes RealtimeTool, a parallel decoding framework for real-time LLM function calling that targets the end-to-end latency bottleneck imposed by autoregressive decoding. The key insight is that structured function-call outputs exhibit substantial redundancy and weak causal dependencies, which the authors exploit via a synergistic design combining special tokens and parallel decoding. Specifically, RealtimeTool introduces special tokens that both compress predictable low-entropy tokens and serve as mode selectors to enable independent, parallel generation of the function name and arguments while sharing a single prefix KV cache. This design allows end-to-end latency to be dominated by the slowest decoding head rather than the total token count. Experiments on multiple benchmarks and Qwen-series models (0.5B–14B) show consistent 3–6× speedups with competitive or improved accuracy, enabling sub-100 ms function-calling latency on consumer GPUs and demonstrating practical applicability to latency-critical scenarios such as mobile agents and real-time interaction.

**Compliance With Llm Reviewing Policy:**

Affirmed.

**Final Justification:**

My major concern is the practical impact in realistic agent pipelines, and has been resolved by the authors.

**Key Questions For Authors:**

1. How does the proposed method perform in more realistic agent settings where action generation is not the dominant bottleneck?

2. How does the proposed approach compare against constrained decoding baselines implemented in existing inference frameworks, such as SGLang (e.g., xGrammar)?

3. To what extent are the observed performance gains attributable to the training strategy versus increased model capacity introduced by LoRA?

**Limitations:**

Yes.

**Strengths And Weaknesses:**

The paper is clearly written and well presented. The overall logic is easy to follow, and the motivation is well supported by concrete latency measurements and empirical observations. The figures and tables effectively illustrate where autoregressive decoding becomes a bottleneck for real-time function calling, making the problem setting and design choices easy to understand.
In addition, the proposed approach achieves substantial improvements in action generation latency. By exploiting the structural properties of function-call outputs and enabling parallel decoding, the method significantly reduces end-to-end tool invocation delay, which is the primary performance metric in latency-critical agent scenarios. The reported speedups are consistent across model scales, indicating that the method is practically effective within the target setting.

Despite these strengths, several aspects limit the perceived novelty and general applicability of the work. Clarifying and addressing the issues below would help strengthen the paper and could improve its overall evaluation.

First, the practical impact of the proposed optimization may be limited in more realistic agent pipelines. Many modern agents follow a ReAct-style paradigm, where reasoning tokens constitute a large fraction of the overall latency, and the execution time of tools themselves can also dominate end-to-end response time[1,2]. In such settings, action generation is no longer the primary bottleneck. While the BFCL benchmark used in this paper emphasizes scenarios with short reasoning traces and fast tool execution, it remains unclear how much benefit the proposed method would provide on more challenging agent benchmarks (e.g., GAIA[3]), where reasoning overhead is high and tool execution is slow, potentially reducing the relative gain from faster action decoding.

Second, the experimental evaluation could be strengthened by more comprehensive baseline comparisons and analysis. In particular, the paper does not compare against constrained decoding methods implemented in existing inference frameworks such as SGLang (e.g., xGrammar), which already exploit structural constraints to accelerate function-call generation without parallel decoding. Such a comparison is important to isolate the added benefit of the proposed approach over strong decoding-level baselines. Moreover, while detailed ablations may be costly, it would be informative to better disentangle the contribution of different design elements (e.g., delimiter optimization versus parameter name generation) to the overall speedup.

Third, it is somewhat unclear how much of the observed performance improvement comes from the proposed training strategy itself versus increased model capacity introduced by a relatively high LoRA rank. Without additional controls (e.g., smaller-rank LoRA or parameter-matched baselines), it is difficult to attribute the gains solely to the parallel decoding design rather than to additional trainable parameters. Clarifying this distinction would help better understand the source of the improvements.

Overall, I like this paper, and I would be happy to reconsider a higher score if the authors can adequately address the concerns outlined above.

[1] Ye N, Ahuja A, Liargkovas G, et al. Speculative Actions: A Lossless Framework for Faster Agentic Systems[J]. arXiv preprint arXiv:2510.04371, 2025.

[2] Kim S, Moon S, Tabrizi R, et al. An llm compiler for parallel function calling[C]//Forty-first International Conference on Machine Learning. 2024.

[3] Mialon G, Fourrier C, Wolf T, et al. Gaia: a benchmark for general ai assistants[C]//The Twelfth International Conference on Learning Representations. 2023.

---

> ### Author Rebuttal · Authors · 2026-03-30
>
> We sincerely thank you for your thoughtful and encouraging review. We are deeply grateful for your recognition that our paper is **"clearly written and well presented"** with **"motivation well supported by concrete latency measurements"** and that our approach **"achieves substantial improvements in action generation latency"** with **"speedups consistent across model scales."** Inspired by your constructive suggestions, we conducted substantial additional experiments that we believe strengthen the paper.
>
> ## Q1. Practical Impact in Realistic Agent Pipelines
>
> We appreciate this important question and have carefully analyzed all cited works: Speculative Actions (Ye et al., 2025), LLM Compiler (Kim et al., 2024), GAIA (Mialon et al., 2023), AsyncLM, and GhostShell.
>
> These works target general-purpose agentic pipelines (e.g., ReAct-style), where the dominant latency bottlenecks lie in external environment feedback, result processing, and chain-of-thought reasoning—not action generation. These scenarios are also not latency-critical by nature: a conversational agent does not require sub-100ms actions.
>
> RealtimeTool targets a fundamentally different class—**embodied intelligence, game AI, and real-time agents**—where the function call *is* the entire output and must complete within a single control cycle (e.g., 100ms at 10 Hz). There is no reasoning chain or tool execution wait—**action generation latency *is* the bottleneck** and directly determines deployability. We readily acknowledge that our acceleration would provide limited benefit in general agentic pipelines; however, for our target scenarios, it is what makes deployment feasible in the first place.
>
> To substantiate this, we provide:
>
> **(1)** on-device iPhone deployment comparing RT-Qwen-0.5B vs. FunctionGemma:
>
> | Model | Params | P50 E2E | P90 E2E | P95 E2E | Thermal State |
> |---|---|---|---|---|---|
> | FunctionGemma | 270M | 709ms | 1570ms | 1831ms | 78.7% fair, 21.3% serious |
> | RT-Qwen-0.5B | 500M | 528ms | 1018ms | 1145ms | 64.7% nominal, 35.3% fair |
>
> Test: iPhone 17 Pro Max (A19 Pro), llama.cpp (identical binary), INT8 Q8\_0, Mobile Actions (961 samples). Despite having ~2× more parameters, RT-Qwen-0.5B achieves 1.34× faster P50 and 1.54× better P90, with significantly better thermal behavior—fewer decoding steps directly reduce sustained thermal load. These results are conservative: llama.cpp lacks both prefix caching and optimized batching backends (e.g., vLLM/SGLang), both of which naturally benefit RealtimeTool's shared-prefix parallel decoding design.
>
> **(2)** Robotic arm control in NVIDIA Isaac Sim (https://anonymous.4open.science/r/temp_icml-C301/, Fig. 1: Isaac Sim with a Franka robot arm and target objects powered by RealtimeTool).
>
> **(3)** A theoretical upper-bound comparison against constrained decoding, showing a >3× gap (detailed in Q2).
>
> ## Q2. Constrained Decoding Baselines (SGLang/XGrammar)
>
> We carefully investigated SGLang, XGrammar, Outlines, and XGrammar 2 (Li et al., 2026). Outlines and XGrammar are logits-masking frameworks—they ensure output validity but **do not reduce forward passes**. The technique that actually reduces forward passes is **jump-forward decoding** (SGLang, Dec 2023), which skips deterministic positions. Jump-forward has been effectively deprecated in current SGLang (Issue #8954), so we provide a theoretical upper-bound analysis on BFCL-v3:
>
> | Method | Avg Forward Passes | Theoretical Speedup |
> |---|---|---|
> | Vanilla (unconstrained) | 40.7 | 1.00× |
> | Jump-forward (upper bound) | 29.5 | 1.38× |
> | RealtimeTool (bottleneck head) | 9.4 | 4.32× |
>
> Even at its theoretical optimum, jump-forward achieves only 1.38× vs. RealtimeTool's 4.32×—a >3× gap.
>
> ## Q3. LoRA Capacity vs. Parallel Decoding Design
>
> To isolate the contribution of our design from increased model capacity, we trained a baseline with **identical LoRA rank (512) and identical data**:
>
> | Configuration | Output Format | Overall Acc | Func Acc |
> |---|---|---|---|
> | Baseline (no training) | JSON | 83.2% | 91.6% |
> | Vanilla SFT (rank 512) | JSON | 71.3% | 97.0% |
> | RealtimeTool (rank 512) | Special tokens | 86.0% | 99.1% |
>
> With identical parameters and data, Vanilla SFT decreases Overall Acc to 71.3% (Func Acc rises to 97.0%, confirming successful training), while RealtimeTool improves to 86.0%. **Gains stem from the parallel decoding design and structural token reduction, not from incremental capacity.**
>
> A LoRA rank ablation further reveals a clear capacity floor:
>
> | LoRA Rank | Overall Acc |
> |---|---|
> | 16 | 69.2% |
> | 32 | 73.7% |
> | 64 | 85.5% |
> | 512 | 85.3% |
> | 1024 | 87.0% |
>
> Performance plateaus beyond rank 64 (512→1024: +1.7%), confirming that the large rank addresses a capacity requirement for learning 8 independent output modes. We adopt rank 512 as default because it balances accuracy, training cost, and GPU memory utilization—rank 1024 strains single-GPU memory and slows training.
>
> ---
>
> Thank you for your support and feedback!

---

> > ### Author Rebuttal · Reviewer_LAba · 2026-04-01
> >
> > My major concern is the practical impact in realistic agent pipelines, and has been resolved by the authors.

---

> > > ### Author Response · Authors · 2026-04-01
> > >
> > > Thank you very much! Have a nice day!

---

### Official Review · Reviewer_Vuhp · 2026-03-13

**Soundness:** 3
**Presentation:** 2
**Significance:** 2
**Originality:** 3
**Overall Recommendation:** 4
**Confidence:** 3

**Summary:**

The authors propose a framework for faster LLM function calling via parallel decoding of function calls. The authors propose using multiple decoding heads to generate function names and arguments independently using the same KV cache. This allows for a significant reduction in the number of tokens needed to decode vs if a full json is generated for the function call. The authors evaluate on several tool use benchmarks and achieve an order of magnitude of speedup relative to a baseline model at little degradation in task performance

**Compliance With Llm Reviewing Policy:**

Affirmed.

**Final Justification:**

After discussion, I update my score: Overall Recommendation 2->4, Presentation 1->2, Soundness 2->3. Overall the work seems to make an effective contribution, though the presentation would be improved further with more explicit examples of settings where real time control is needed and the current work would meet the need. Most of my concerns are addressed, and my remaining reservations (around generalization to more challenging settings with e.g. more tools) feel fair to leave as out of scope of the current work as long as the limitations are clearly outlined in the updated manuscript

**Key Questions For Authors:**

1. What exactly does training look like? Is the LLM trained to also predict *whether* a tool call is needed in the first place? Or is the training objective set up to assume that the LLM already has predicted that a tool call is needed, merely needs to predict what the tool calls are? Are more/larger GPUs ever required to account for multiple argument heads being trained in parallel?
2. Likewise, what exactly does inference look like? What are hardware requirements for serving a model trained to do parallel decoding with the authors' method -- are more/larger GPUs ever required to account for multiple argument heads being decoded in parallel?

**Limitations:**

1. Authors mention computational overhead in training due to the multiple heads, vs standard fine-tuning, but the authors do not quantify or further characterize the overhead (see above)
2. It would have been helpful to understand what the failure modes are. In particular, are there any intuitions on adversarial or challenge settings where the proposed scheme would break down? I would expect that conditioning on the function name should theoretically be helpful for predicting accurate argument tokens, especially as the space of possible functions grows large.

**Strengths And Weaknesses:**

Strengths:
1. Conceptually simple and clean proposed method aiming to address a latency bottleneck in LLM inference involving function calling
2. The proposed approaches are empirically effective in the settings explored
3. Interesting analysis/approach for addressing argument number imbalance

Weaknesses:
1. Problematic framing around the "real-time gap" premise underlying the work. Missing citation provided for the "sub-100ms threshold for responsive control" mentioned by the authors -- in any case, however, it is not obvious to me that this notion of "responsive control" is relevant to LLM specific latency concerns. Specifically, it could be described in much more detail what the implications of "real-time LLM function calling" might be. Tool calls in conversational applications seem most likely inconsequential, and on the other hand I can imagine that <100ms responsiveness could be helpful or necessary in certain settings currently not as widely used precisely due to the latency gap that the authors aim to address. Appendix A lists "real-time LLM applications" that are relevant to this concern, but **critically, it is not clear to me that any of the benchmarks that the authors *do* evaluate on are obviously realistic wrt a premise of requiring real-time control.**
2. Major issues with presentation and clarity -- see below (list includes both major issues and general in-line comments)
3. Ablation studies are not ablation studies in the traditional sense

In-line comments/weaknesses:
1. "Constrained Decoding" paragraph in intro: to my understanding, SGLang is not itself a "grammar-guided framework" the same way that XGrammar and Outlines are -- in fact, SGLang supports XGrammar and Outlines as a backend for structured decoding
2. I would strongly prefer to see Figure 1 on page 2 if the abstract does not fit on the first page
3. Incorrect in-text citations for BFCL (Patil 2025) -- usually linked to Gorilla (Patail 2024) instead
4. I felt that a related work paragraph on token compression and pruning would have been appropriate
5. 2.1.3: "8-head parallel decoding achieves 93.0% average TPOT efficiency" -- "TPOT efficiency" should be clearly defined before use, and the definition should be in the main body vs. unlinked in the Appendix (G.1).
6. 3.1: p50 and p90 should probably be mentioned as well in the "metrics" paragraph
7. Line 397 in 3.5: why 64 lora rank in the 3.4 case study, if 512 is adopted as a "default"?
8. Unclear from methods section what the training methods looked like -- I can infer some information from the case study in 3.4 (it seems like training is on the respective training sets for each dataset?), but this was not enough information for me to understand what exactly was done to train the models

---

> ### Author Rebuttal · Authors · 2026-03-30
>
> We thank you for your detailed review and recognition that our method is "conceptually simple and clean" and "empirically effective." Your in-line suggestions on presentation have been particularly valuable. Here are our clarifications and supplementary experiments.
>
> ## Q1. "Real-Time Gap" Premise and Benchmark Relevance
>
> RealtimeTool targets applications where the LLM serves as a real-time controller, not a conversational reasoner, where the function call is the entire model output and latency directly determines whether the method is deployable.
>
> The sub-100ms threshold is well-supported: autonomous driving requires ~100ms e2e latency (Luo et al., 2019); VLA control loops need ≥10 Hz for stability (ActionFlow, 2024).
>
> Our benchmarks validate accuracy across diverse sources, while latency is task-agnostic—the decoding cost of a valid function call is equivalent whether it controls a robotic arm or invokes a weather API. To substantiate these claims beyond benchmarks, while no established public benchmark exists for evaluating LLMs as real-time controllers, we provide: (1) **on-device iPhone deployment** with apple-to-apple RT-Qwen-0.5B vs. FunctionGemma comparison (details in response to Reviewer LAba, Table 1); (2) **robotic arm control in NVIDIA Isaac Sim** (https://anonymous.4open.science/r/temp_icml-C301/, Fig. 1: Franka robot arm powered by RealtimeTool).
>
> ## Q2. Presentation Issues
>
> We will address all identified issues in revision:
>
> - **SGLang classification**: Upon closer investigation, we note that Outlines and XGrammar are logits-masking frameworks—they improve output validity but do not reduce the number of forward passes. The technique that reduces e2e latency is jump-forward decoding (SGLang, Dec 2023). SGLang integrates these components together in practice, which led to our imprecise grouping. This will be corrected.
> - **BFCL citation**: We follow the official BibTeX (https://gorilla.cs.berkeley.edu/leaderboard.html). Concurrent works such as XGrammar 2 (Li et al., 2026) also cite the 2025 version.
> - **Other fixes**: Figure 1 → page 2; token compression/pruning related work added; TPOT efficiency defined in main body before first use; P50/P90 introduced in Section 3.1 metrics.
> - **LoRA rank=64 in 3.4**: Section 3.4 adapts an already-trained RT-Qwen-0.5B, not the baseline—hence the smaller rank. Will be clarified.
>
> ## Q3. Ablation Studies
>
> We note that compression and parallel decoding are synergistic. Removing one eliminates the precondition for the other. However, we can quantify each mechanism's contribution:
>
> **(1) Compression:** Over 4× token reduction (Table 1 in paper). We supplement with a forward-pass comparison against the theoretical upper bound of jump-forward decoding on BFCL-v3: 4.32× vs. 1.38× vs. 1.00× for RealtimeTool, Jump-forward, and Vanilla. (Table 1 in response to Reviewer k8dV)
>
> **(2) Parallelization overhead:** Only +8.2% for 8-head decoding (Appendix G), near-free parallelism.
>
> **(3) LoRA rank:** We acknowledge that choosing rank 512 as default when rank 1024 achieves best accuracy was insufficiently explained. The ablation we want to demonstrate is that small ranks cannot adequately train 8 independent output modes. We choose rank 512 to achieve a balance between model quality and resource utilization.
>
> ## Q4. Training and Inference
>
> **Training:** The model learns both *whether* and *what* to call (\~3% additional zero-argument function `pass()` samples for tool-call gating). We train on a unified dataset and evaluate across all benchmarks without benchmark-specific tuning. We break the original function calling data into 8 heads, where each head is processed as an independent sequence, increasing training time (~5×) but no additional memory, details shown in Appendix F. Domain adaptation costs much lower, as shown in Case Study 3.4. The additional training overhead can be further reduced by training prefix sharing (Wang & Hegde, 2024).
>
> **Inference:** No additional GPUs or memory required—parallel decoding exploits idle compute during the memory-bandwidth-bound decode phase. We also design a three-level KV cache hierarchy as described in Appendix E.1.4 (details in response to Reviewer UYJK, A2).
>
> ## Q5. Failure Modes (see also in response to Reviewer k8dV, Q5)
>
> Regarding the concern about scaling to large function spaces: across our benchmarks, 25.3% of samples involve >7 candidate tools, yet RealtimeTool maintains 99.1% average function accuracy.
>
> We sincerely appreciate the reviewer's suggestion that conditioning on the function name could help predict argument tokens more accurately. This is a perceptive and constructive insight—we explored such conditional designs in early experiments. However, this would introduce a sequential dependency that adds decode steps to the critical path, conflicting with our core goal of minimal latency. For scenarios with less stringent latency requirements, this is a promising direction worth pursuing.
>
> Thanks for your feedback!

---

> > ### Author Rebuttal · Reviewer_Vuhp · 2026-04-04
> >
> > Thank you for your engagement!
> >
> > > BFCL citation
> >
> > I am still not clear on why sometimes BFCL is cited as the 2024 version and it is sometimes cited as the 2025 version. Is it just v3 vs v1?
> >
> > > Appendix F
> >
> > Appendix F is only hyperparameters -- it would be helpful to add details of the actual training procedure and objective a la what is described in the rebuttal
> >
> > > 7 candidate tools
> >
> > 7 candidate tools is still not very many to my understanding. I did not expect the authors to pull new empirical results out of thin air on this, but I would like to see explicit engagement with at least the hypothetical of what would happen "as the space of possible functions grows large" -- my concern is that the 99.1% is predicated on there being only a small number of candidate tools.

---

> > > ### Author Response · Authors · 2026-04-04
> > >
> > > Thank you for the continued engagement and these focused follow-up questions.
> > >
> > > ## BFCL Citation
> > >
> > > These are **two distinct papers** by overlapping author groups:
> > >
> > > - **Patil et al., 2024** = *"Gorilla: Large Language Model Connected with Massive APIs"* (NeurIPS 2024), which introduced the Gorilla model and the OpenFunction benchmark.
> > > - **Patil et al., 2025** = *"The Berkeley Function Calling Leaderboard"* (ICML 2025), which introduced the BFCL leaderboard and the v3 evaluation protocol.
> > >
> > > We cite the former when referring to OpenFunction-v1 and the latter when referring to BFCL-v3. We will add explicit disambiguation in the revision.
> > >
> > > ## Appendix F
> > >
> > > We fully agree. The detailed training procedure described in our rebuttal (Q4) will be incorporated into Appendix F in the revision, covering data organization into 8 parallel head sequences, the per-head next-token prediction objective with head-specific loss weighting, tool-call gating via pass-through samples, and the practical training flow. If page budget permits, we will also consider moving key training details into the main body to improve self-containedness. We sincerely apologize for the writing oversight and the confusion it caused.
> > >
> > > ## Large Function Space
> > >
> > > We appreciate this question. First, a clarification: the threshold of 7 tools was chosen because Mobile Actions — the real-world benchmark that Google selected for evaluating FunctionGemma — defines exactly 7 candidate tools.
> > >
> > > We provide a stratified analysis of Qwen3-4B on our full evaluation suite (5,294 samples):
> > >
> > > | # Tools | Samples | RT Overall | RT Func Acc | Baseline Overall | Baseline Func Acc |
> > > |---------|---------|------------|-------------|-----------------|-------------------|
> > > | 1       | 1,145   | 85.6%      | **99.9%**   | 82.3%           | 94.8%             |
> > > | 2       | 563     | 84.0%      | **98.8%**   | 82.2%           | 94.7%             |
> > > | 3       | 688     | 84.5%      | **97.8%**   | 82.4%           | 93.0%             |
> > > | 4       | 488     | 85.3%      | **98.0%**   | 85.3%           | 93.2%             |
> > > | 5       | 426     | 85.2%      | **97.9%**   | 80.8%           | 92.0%             |
> > > | 6       | 650     | 89.5%      | **99.7%**   | 85.5%           | 93.1%             |
> > > | 7       | 1,310   | 84.4%      | **99.9%**   | 68.7%           | 75.3%             |
> > > | 9       | 11      | 81.8%      | 90.9%       | 90.9%           | 100.0%            |
> > > | 10      | 12      | 100.0%     | 100.0%      | 100.0%          | 100.0%            |
> > > | 26      | 1       | 100.0%     | 100.0%      | 100.0%          | 100.0%            |
> > >
> > > Samples with ≥9 tools are too few for statistical conclusions, but are included for completeness.
> > >
> > > RealtimeTool maintains stable function accuracy (97.8–99.9%) from 1 to 7 tools with no observable degradation. The baseline, by contrast, drops sharply at 7 tools (Func Acc: 93% → 75.3%). This is likely because the baseline must decode substantially more tokens per function call, which increases the probability of generation errors — particularly for the small model scales (0.5B–4B) targeted by our work.
> > >
> > > We acknowledge that our benchmarks do not cover scenarios with hundreds of candidate tools. As the tool space grows very large, we expect challenges to arise primarily on the prompt side — fitting tool definitions into the context window, maintaining attention quality, and discriminability of tool descriptions. These challenges would pose a high risk of accuracy degradation for small models regardless of the decoding strategy, and affect sequential and parallel decoding equally. In practice, methods such as RAG-based pre-filtering are typically employed to keep the candidate set manageable before generation. We thank you for raising this important point and will add this discussion to the limitations section.
> > >
> > > We sincerely thank you again for engagement throughout the review process. Should any further questions arise, we warmly welcome continued discussion.

---

### Official Review · Reviewer_k8dV · 2026-03-14

**Soundness:** 2
**Presentation:** 3
**Significance:** 3
**Originality:** 3
**Overall Recommendation:** 4
**Confidence:** 3

**Summary:**

The paper proposes a function-calling framework, RealtimeTool, that speeds up LLM tool use by compressing structured outputs with special tokens, and then decoding the function name and arguments in parallel heads that share the same prefix KV cache.

The key idea is that tool calls contain large amounts of low-entropy formatting plus relatively weak dependencies across arguments, so compression and parallel decoding should be designed together. Experiments on Qwen models across five benchmarks report roughly 4-6x reduction in token count and 3-6x e2e speedup.

**Compliance With Llm Reviewing Policy:**

Affirmed.

**Final Justification:**

The paper offers an interesting perspective that brings parallelization to tool using. The reviewer thinks the paper would benefit if its scope can be sharpened and its training setup further clarified. As written, it mainly improves tool use function-calling latency, which does not yet make the practical benefit super convincing given agents’ high e2e latency in many agentic applications involve other forms of text generations beyond function calling. Maybe the authors can consider narrow down the scope to physcial ai or specific edge applications.

**Key Questions For Authors:**

1. Can you add direct comparisons against at least one structured-generation baseline such as SGLang/Outlines/XGrammar, and one parallel tool-calling baseline such as LLMCompiler or AsyncLM? This would materially clarify the paper’s contribution for practical use.
2. Why is rank 512 chosen as the default when the ablation reports the best BFCL-v3 accuracy at rank 1024? A comprehensive speed-quality (with training cost annotations) tradeoff would help.

**Limitations:**

The paper has limitations section, covering argument independence, fixed head count, training cost, along with an impact statement that mentioned risks from autonomous deployment without oversight.

**Strengths And Weaknesses:**

Strengths
1. Overall, I like the idea because it is intuitive and effective: it leverages the low-entropy tokens in structured generation to enable parallel decoding.
2. The empirical results are strong on latency improvement and reasonably broad exp settings on six Qwen-family backbones, five tool-calling benchmarks, two hardware classes (showing both vLLM and hf implementations, which is solid), and a quantized deployment setting.
3. The FunctionGemma comparison is useful and strengthens the edge-deployment story: the paper shows RT-Qwen-0.5B beating Google’s 270M FunctionGemma on Mobile Actions in both accuracy and latency.

Weaknesses
1. While evaluation settings are broad, the baseline set is incomplete for the paper’s claim. In the main experiments, the comparison is mostly against vanilla Qwen baselines plus FunctionGemma on one benchmark, while several close practical baselines discussed in related work: (1) SGL/Outlines/XGrammar for structured generation and (2) LLMCompiler/AsyncLM/GhostShell for parallel or concurrent tool execution, are not evaluated directly. That makes the “real-time function calling” claim less convincing without showing consistent comparative advantages. Adding those baselines would make the paper stronger.
2. The method depends on a fairly specific setup: it assumes argument independence, and requires relatively high-capacity LoRA plus synthetic augmentation to work well. Also, the default LoRA rank is not the best ablation setting shown. The paper would also benefit from more comprehensive ablations on the design choices and failure regimes.

---

> ### Author Rebuttal · Authors · 2026-03-30
>
> We sincerely thank you for your thoughtful and encouraging review. We greatly appreciate your recognition that our idea **"is intuitive and effective: it leverages the low-entropy tokens in structured generation to enable parallel decoding."** We are also grateful for your positive assessment that our **"empirical results are strong on latency improvement and reasonably broad"**. Furthermore, we value your acknowledgment that our **"FunctionGemma comparison is useful and strengthens the edge-deployment story."** Inspired by your constructive suggestions, we conducted substantial additional experiments that we believe significantly strengthen the paper.
>
> ## Q1. Constrained Decoding Baselines (SGLang/Outlines/XGrammar)
>
> We note that Outlines and XGrammar are logits masking frameworks—they constrain outputs to valid formats but do not reduce forward passes; masking introduces additional overhead. The mechanism that reduces forward passes is **jump-forward decoding** (SGLang, Dec 2023), which skips deterministic positions. Despite significant effort,  we found jump-forward has been effectively deprecated in current SGLang (Issue #8954), making reliable benchmarking infeasible. We provide a theoretical upper-bound analysis on BFCL-v3 :
>
> | Method | Avg Forward Passes | Theoretical Speedup |
> |---|---|---|
> | Vanilla (unconstrained) | 40.7 | 1.00× |
> | Jump-forward (theoretical upper bound) | 29.5 | 1.38× |
> | RealtimeTool (bottleneck head) | 9.4 | 4.32× |
>
> Even at its theoretical optimum, jump-forward achieves only 1.38× vs. RealtimeTool's 4.32×—a >3× gap. In practice, the actual speedup would be further diminished by engineering overhead, making this upper bound unattainable for real-world function calling workloads.
>
> ## Q2. LLMCompiler/AsyncLM/GhostShell
>
> They operate at different levels. LLMCompiler and AsyncLM are agentic scheduling optimizations where the bottleneck lies in tool execution and result processing—not in generating individual tool calls. GhostShell enables streaming-based early parsing but remains autoregressive, reducing zero forward passes. Our target is where tool-call generation itself is the bottleneck—e.g., real-time robotic control, mobile actions. We provide:
>
> (1) a robotic arm demo in Isaac Sim powered by RealtimeTool (https://anonymous.4open.science/r/temp_icml-C301/, Fig. 1: Isaac Sim with a Franka robot arm and target objects powered by RealtimeTool);
>
> (2) on-device apple-to-apple iPhone testing where RT-Qwen-0.5B outperforms FunctionGemma in both latency and thermal behavior (detailed in response to Reviewer LAba, Table 1).
>
> ## Q3. LoRA Rank Selection
>
> We acknowledge that **"512 chosen as the default when the ablation reports best accuracy at rank 1024"** was insufficiently explained. The goal of ablation is to demonstrate that small ranks cannot adequately train 8 independent output modes—a capacity floor. We supplement with ranks 16/32:
>
> | LoRA Rank | Overall Acc |
> |---|---|
> | 16 | 69.2% |
> | 32 | 73.7% |
> | 64 | 85.5% |
> | 512 | 85.3% |
> | 1024 | 87.0% |
>
> A clear cliff exists below 64; beyond that, gains plateau (512→1024: +1.7%). Rank 512 was chosen because: (1) it is within the plateau; (2) rank 1024 strains single-GPU memory and slows training; (3) rank 512 happens to fully utilize our GPU memory budget.
>
> ## Q4. Synthetic Data and LoRA Capacity
>
> Synthetic augmentation balances argument-count distributions for multi-head learning—not a general quality enhancement. To isolate gains, we trained baselines with the same data and LoRA rank 512:
>
> | Configuration | Format | Overall Acc | Func Acc |
> |---|---|---|---|
> | Baseline (no training) | JSON | 83.2% | 91.6% |
> | Vanilla SFT | JSON | 71.3% | 97.0% |
> | RealtimeTool | Parallel Decoding | 86.0% | 99.1% |
>
> Vanilla SFT decreases Overall Acc to 71.3% (Func Acc rises, confirming successful training), while RealtimeTool improves to 86.0%. Gains stem from the parallel decoding design and the reduction of structural tokens, not from incremental data or parameters.
>
> ## Q5. Training Cost and Failure Modes
>
> Training costs ~5× vanilla SFT H100-hours, with no extra memory. Prefix sharing during training (Wang & Hegde, 2024) could reduce this to near-vanilla levels. We believe this one-time cost is justified by the inference speedup; subsequent adaptation (Section 3.4) incurs much lower cost.
>
> Failure modes (see also Appendix I.3):
>
> (1) **Long single-argument** values reduce speedup as the bottleneck head dominates;
>
> (2) **Causal dependencies** between arguments—rare in well-designed APIs though occasionally present in benchmarks, easily addressed by consolidating dependent parameters into a single head;
>
> (3) **A large number of similar tools** may challenge function name and parameters prediction—addressable via prompt design;
>
> (4) APIs with **>6 parameters** require sequential overflow handling (6 heads cover >95% of benchmarks, and the head count can also be reduced to fewer than 6 for simpler schemas).
>
> Thanks for your suggestions and support!

---

> > ### Author Rebuttal · Reviewer_k8dV · 2026-04-03
> >
> > Thank the authors for additional clarifications and experiments. My previous questions have been fully addressed. Upon cross referencing other reviews, I would like to ask additional question about the benchmark choices: are there specific reasons why the authors choose weather API tasks and robotic arm control (which is more of a embodied AI task)? Can RealtimeTool be applied to other agentic tasks like coding and web browsing with practical performance gains?

---

> > > ### Author Response · Authors · 2026-04-04
> > >
> > > Thank you for the positive acknowledgement and for raising this thoughtful follow-up question.
> > >
> > > We appreciate the opportunity to provide more context on our benchmark and domain coverage. The weather API (`get_weather`) in Figures 2–3 serves as a running illustrative example for readability, and the robotic arm demo was provided during discussion to demonstrate real-time deployment feasibility. Our evaluation is in fact conducted on diverse, general-purpose function calling benchmarks spanning a broad range of domains:
> > >
> > > - **BFCL-v3** (the de facto standard for function calling evaluation): the Live subsets contain real-world user-contributed APIs from domains such as ridesharing, GitHub analytics, payments, and home automation; the Executable subsets verify actual API execution across scenarios like file systems and database queries; and the Non-Live subsets cover expert-curated functions across Python, Java, and JavaScript.
> > > - **SealTools** contains 4,076 automatically generated APIs across various life domains, including nested tool calling scenarios.
> > > - **ToolAlpaca** spans 50 categories with 271 tool-use instances.
> > > - **OpenFunction** covers 65 distinct APIs.
> > > - **Mobile Actions** targets 7 smartphone operations for mobile device control.
> > >
> > > Our training data (xLAM-60K) is similarly broad, built from 3,673 executable APIs across **21 categories** including finance, social media, education, mathematics, sports, technology, travel, health, and others.
> > >
> > > Notably, all main results (Table 2) are obtained from a single general-purpose model per backbone — we do not perform any benchmark-specific or domain-specific adaptation. The model learns multi-head parallel generation as a general capability. Domain-specific adaptation is only explored in Case Study 3.4, where we fine-tune on the Mobile Actions training split to compare with FunctionGemma.
> > >
> > > Regarding applicability to other agentic tasks such as coding and web browsing: our acceleration operates at the decoding level — it reduces the number of forward passes required to generate any structurally valid function call, independent of the application domain. In agentic workflows that involve frequent function calling turns (e.g., multi-step pipelines where many sub-agents each issue tool calls), RealtimeTool would accelerate every such generation step, and the cumulative savings could be substantial. That said, we acknowledge that in settings where the dominant latency comes from external tool execution or extended reasoning, the relative contribution of our acceleration to total end-to-end time would be diluted accordingly.
> > >
> > > We sincerely thank you for your continued engagement and discussion throughout the review process. Should any further questions arise, we warmly welcome continued discussion.

---

### Decision · Program_Chairs · 2026-04-30

**Decision:**

Accept (regular)

**Comment:**

This paper presents a parallel decoding framework for LLM function calling that compresses structural tokens and generates function arguments in parallel via multi-head decoding, achieving 3–6× speedup.
Three reviewers rate Weak Accept (4) and one Weak Reject (3). Reviewers consistently praise the clean, intuitive design, strong latency results across model scales, and comprehensive benchmarking including on-device iPhone and robotic arm demonstrations.
The central debate is scope and practical impact. Multiple reviewers question whether accelerating only the function-call generation step matters in broader agentic pipelines where reasoning and tool execution dominate latency.
The contribution is narrow but well-executed for its target setting. Missing comparisons against general acceleration methods (speculative decoding, MTP) as direct baselines remain a minor gap, though the authors discuss complementarity.